



# Alkalinity generation from carbonate weathering in a silicate-dominated headwater catchment at Iskorasfjellet, northern Norway

Nele Lehmann[1,2,3], Hugues Lantuit[2,4], Michael Ernst Böttcher[5,6,7], Jens Hartmann[8], Antje Eulenburg[2], Helmuth Thomas[1,3]

[1]Institute of Carbon Cycles, Helmholtz-Zentrum Hereon, Geesthacht, D-21502, Germany
[2]Alfred Wegener Institute Helmholtz Centre for Polar and Marine Research, Potsdam, D-14473, Germany
[3]Institute for Chemistry and Biology of the Marine Environment (ICBM), University of Oldenburg, Oldenburg, D-26129, Germany
[4]Institute for Geosciences, University of Potsdam, Potsdam, D-14476, Germany
[5]Geochemistry and Isotope Biogeochemistry, Leibniz Institute for Baltic Sea Research (IOW), Warnemünde, D-18119, Germany
[6] Marine Geochemistry, University of Greifswald, Greifswald, D-17489, Germany
[7]Interdisciplinary Faculty, University of Rostock, Rostock, D-18059, Germany
[8]Institute for Geology, Center for Earth System Research and Sustainability, University Hamburg, Hamburg, D-20146, Germany

*Correspondence to*: Nele Lehmann (nele.lehmann@hereon.de) and Helmuth Thomas (helmuth.thomas@hereon.de)

**Abstract.** The weathering rate of carbonate minerals is several orders of magnitude higher than for silicate minerals. Therefore, small amounts of carbonate minerals have the potential to control the dissolved weathering loads in silicate-dominated catchments. Both weathering processes produce alkalinity under the consumption of $CO_2$. Given that only alkalinity generation from silicate weathering is thought to be a long-term sink for $CO_2$, a misattributed weathering source could lead to incorrect conclusions about long- and short-term $CO_2$ fixation. In this study, we aimed to identify the weathering sources responsible for alkalinity generation and $CO_2$ fixation across watershed scales in a degrading permafrost landscape in northern Norway, 68.7 – 70.5 °N, and on a temporal scale, in a subarctic headwater catchment on the mountainside of Iskorasfjellet, characterized by sporadic permafrost and underlain mainly by silicates as the alkalinity-bearing lithology. By analysing total alkalinity (AT) and dissolved inorganic carbon (DIC) concentrations, as well as the stable isotope signature of the latter ($\delta^{13}$C-DIC) in conjunction with dissolved cation and anion loads, we found that AT was almost entirely derived from weathering of the sparse carbonate minerals. We propose that in the headwater catchment, the riparian zone is a hotspot area of AT generation and release due to its enhanced hydrological connectivity, and that the weathering load contribution from the uphill catchment is limited by insufficient contact time of weathering agent and weatherable material. By using stable water isotopes, it was possible to explain temporal variations in AT concentrations following a precipitation event due to surface runoff. In addition to carbonic acid, sulphuric acid, probably originating from pyrite oxidation, is shown to be a potential corrosive reactant. An increased proportion of sulphuric acid as a potential weathering agent may have resulted in a decrease in AT. Therefore, carbonate weathering in the studied area should be considered not only as a short-term $CO_2$ sink, but also as a potential $CO_2$ source. Finally, we found that AT increased with decreasing permafrost probability, and attributed this relation to an increased





water storage capacity associated with increasing contact of weathering agent and rock surfaces, and enhanced microbial activity. As both soil respiration and permafrost thaw are expected to increase with climate change, increasing the availability of weathering agent in the form of $CO_2$ and water storage capacity, respectively, we suggest that future weathering rates and alkalinity generation will increase concomitantly in the study area.

**1 Introduction**

Weathering of silicate rocks is thought to be the long-term sink (millions of years) for atmospheric $CO_2$ (Berner et al., 1983; Garrels and Berner, 1983). While alkalinity generation from weathering of calcium-silicates contributes to the long-term drawdown of $CO_2$ via the precipitation of carbonates in the ocean and thus the return of carbon to the lithosphere, the alkalinity generation from weathering of carbonate rocks does not have such an impact on these timescales. Carbonate weathering is

thought to in equilibrium with marine calcification over shorter time-scales (<10,000 years; Zeebe and Westbroek (2003)). Global estimates (Gaillardet et al., 1999; Amiotte Suchet et al., 2003; Hartmann et al., 2009) attribute 49-63% of $CO_2$ consumption by terrestrial weathering to silicate rocks, the remainder to carbonate rocks. These studies rely on lithological maps and bulk river chemistry data used to infer whether $CO_2$ consumption is due to silicate or carbonate weathering. However, some case and regional studies (Blum et al., 1998; White et al., 1999; Jacobson et al., 2002; Jacobson et al., 2003; Oliver et

al., 2003; White et al., 2005; Moore et al., 2013; Jacobson et al., 2015) indicate that the global calculations relying on bulk river chemistry data may overestimate silicate weathering. Hartmann (2009) pointed out that interpretation of the molar ratio of $Ca^{2+}/Na^+$ may lead to incorrect attribution of $CO_2$ drawdown to silicate weathering in predominantly silicate areas, which should instead be attributed to the weathering of accessory carbonate minerals. He calculated trace carbonate contribution for igneous rocks and silicate sediments, and applied the developed method globally (Hartmann et al., 2009). Especially silicate-

dominated regions which are physically active or show early stages of weathering (e.g. during and following periods of glaciation and tectonism) show high weathering loads of accessory carbonate minerals (Jacobson et al., 2002; Jacobson et al., 2003; White et al., 1999; Oliver et al., 2003; White et al., 2005; Moore et al., 2013; Jacobson et al., 2015). The weathering rate of carbonates is several orders of magnitude faster than the one of silicates (Lasaga, 1984; Stallard and Edmond, 1987; Jacobson et al., 2003), thus dissolved inorganic carbon (DIC) is often controlled by carbonates (Liu et al., 2018).


DIC is composed of $CO_2^*$ (i.e. the sum of dissolved $CO_2$ and $H_2CO_3$), $HCO_3^-$ and $CO_3^{2-}$, with the relative proportion of the DIC species depending on alkalinity and environmental conditions such as temperature. In fresh water, total alkalinity (AT), or acid-neutralizing capacity, is mainly composed of carbonate alkalinity (i.e. $HCO_3^-$ and $CO_3^{2-}$). At pH values between 7 and 9, the alkalinity concentration is approximately equal to the $HCO_3^-$ concentration (~95% of the carbon in the water is in the

form of $HCO_3^-$), as the equilibrium between $CO_2^*$, $HCO_3^-$ and $CO_3^{2-}$ in this pH range is strongly in favour of $HCO_3^-$ (Stumm and Morgan, 1981). Alkalinity in the form of $HCO_3^-$ is produced from the weathering of carbonate minerals (shown here for calcite) under the consumption of one equivalent of atmospheric and/or soil $CO_2$ as follows:



$$CaCO_3 + CO_2 + H_2O \leftrightarrow Ca^{2+} + 2\, HCO_3^- \tag{1}$$

Assuming that the weathering agent is not carbonic acid, but other inorganic acids such as sulphuric and nitric acid, carbonate weathering (shown here for calcite weathering with sulphuric acid) releases $CO_2$, thereby increasing the DIC concentration, but not contributing to alkalinity generation (Berner and Berner, 1987; Marx et al., 2017a; Liu et al., 2018):

$$CaCO_3 + H_2SO_4 \leftrightarrow Ca^{2+} + SO_4^{2-} + H_2O + CO_2 \tag{2}$$

Sulphuric acid can be generated from the dissolution of sulphate-containing minerals, such as gypsum, or the oxidation of sulphide minerals, such as pyrite. Furthermore, both sulphuric and nitric acid can be brought into the soil via acid rain. Finally, nitric acid can also be produced from the oxidation of ammonium fertilizers (Marx et al., 2017a; Li et al., 2010).

While in carbonic acid induced carbonate weathering only of the two equivalents of $HCO_3^-$ is derived from atmospheric and/or soil $CO_2$, in silicate weathering all $HCO_3^-$ originates from $CO_2$ (shown here for anorthite weathering, one of the three major types of feldspar:

$$CaAl_2Si_2O_8 + 2\, CO_2 + 3\, H_2O \rightarrow Al_2Si_2O_5(OH)_4 + Ca^{2+} + 2\, HCO_3^- \tag{3}$$

When silicates are weathered by non-carbon based acids, neither DIC nor AT is generated (shown here for anorthite weathering with sulphuric acid):

$$CaAl_2Si_2O_8 + H_2SO_4 \rightarrow 2\, AlOOH + Ca^{2+} + SO_4^{2-} + 2\, SiO_2 \tag{4}$$

A valuable tool to distinguish between the sources of DIC in streams is the stable isotope composition of DIC, $\delta^{13}C$-DIC (Deines et al., 1974; Böttcher, 1999), which varies across a wide range, typically from +5‰ to -35‰ (Campeau et al., 2017). In addition to the above mentioned geogenic sources, DIC is also governed by biogenic sources, $CO_2$ evasion, and in-stream processes (Kempe, 1982; Campeau et al., 2017). Biogenic DIC originates from autotrophic respiration or organic matter mineralization. In regions dominated by C3 plant vegetation, this biogenic DIC has a typical $\delta^{13}C$ value of about -27‰ (O'Leary, 1988). When measured in soil solution, this value typically increases by 1-4‰, as dissolution and gas exchange across the soil-atmosphere interphase take place (Cerling et al., 1991; Davidson, 1995; Amundson et al., 1998). When carbonate minerals, which have a typical $\delta^{13}C$ value of about 0‰ (Hoefs, 1973; Land, 1980), are weathered by soil respired dissolved $CO_2$ (about -24‰), the final DIC is at saturation with carbonate minerals is characterized by an isotopic composition of about -12‰, considering evolution under conditions closed with reference to a $CO_2$ gas phase in C3 vegetation-dominated catchments (Deines et al., 1974). C4-type of vegetation leads to shift towards heavier stable isotope values (Deines et al., 1974). When the weathering takes place by other inorganic acids, the resulting $\delta^{13}C$-DIC can even turn more positive (Schaefer and Usdowski, 1987, 1992; Michaelis, 1992) and reach under extreme conditions the $\delta^{13}C$ value of carbonate minerals of about 0‰ (Lehn et al., 2017). The actual isotope composition of DIC depends also on the boundary conditions (presence of biogenic $CO_2$) during the groundwater evolution (Deines et al., 1974; Schaefer and Usdowski, 1987, 1992; Böttcher, 1999). DIC generated from carbonic acid induced silicate weathering has a typical $\delta^{13}C$ value of about -24‰, as soil $CO_2$ is the dominant source for DIC (Lehn et al., 2017; Purkamo et al., 2022) and fractionation between the aqueous and gas phase is small at low pH values (Deines et al., 1974).



When $CO_2$ is degassed from the stream due to supersaturation, which is especially prominent in headwater catchments (Michaelis et al., 1985; Marx et al., 2017a), the remaining stream water DIC is enriched in $^{13}C$ (Michaelis et al., 1985; Liu and Han, 2020). Equilibrium exchange fractionation between stream DIC and atmospheric $CO_2$, which has a typical $\delta^{13}C$ value of about -8‰ (Trolier et al., 1996), has the same effect on stream $\delta^{13}C$-DIC as $CO_2$ outgassing (Michaelis et al., 1985; Liu and Han, 2020).


Finally, the $\delta^{13}C$-DIC values on flowing surface waters may also be influenced by in situ biogeochemical processes such as biological respiration, DOC photo-oxidation, photosynthesis, mixing with groundwaters, and anaerobic metabolism (Campeau et al., 2017).


The general controls on chemical weathering are the availability of weatherable minerals, mainly provided by physical weathering of regolith, and the supply of weathering agents, i.e. acids (Raymond and Hamilton, 2018). Further, to generate the soluble weathering products, favorable conditions such as high temperatures, abundant moisture and high contact of mineral surfaces with water should prevail. Lastly, the hydrological transport of these solutes out of the weathering zone is another controlling factor (Raymond and Hamilton, 2018).


Feedback between the Earth's carbon cycle and terrestrial weathering was originally thought to be slow, and $CO_2$ consumption by terrestrial weathering to be at steady-state since pre-industrial times (Walker et al., 1981; Berner et al., 1983). However, a global study (Goll et al., 2014) reported increased $CO_2$ consumption since 1850, and regional studies (Li et al., 2008; Gislason et al., 2009; Raymond et al., 2008; Drake et al., 2018; Macpherson et al., 2019) found an increase in riverine alkalinity over the last decades and related this to changes in temperature, precipitation, vegetation, availability of acids, liming, or hydrologic


flow conditions. Thawing permafrost in cold regions was assumed, too; thereby advocating for a possibly rapid (decadal) feedback between climate and land-use change and riverine alkalinity generation upon terrestrial weathering. Especially carbonate weathering was found to be very responsive to contemporary environmental changes (Michaelis, 1992; Zeng et al., 2019). Besides the tropical region, northern high latitudes are expected in the future to experience enhanced carbonate weathering and thus a higher carbon-sink function due to increased soil $CO_2$ partial pressures and temperatures (Zeng et al.,


2022). The aquifers and soils of Fennoscandia, however, are mainly composed of non-carbonate rocks (O'Nions et al., 1970; Hartmann and Moosdorf, 2012; Zeng et al., 2022), and therefore, a response in terms of $CO_2$ sequestration through weathering on environmental changes might be expected to be slow (Moosdorf et al., 2011).

Climate change is particularly accelerated in the Arctic, as reflected in the annually averaged near-surface air temperature, which increased by 0.71°C per decade from 1979 to 2021, nearly four times faster than the global average (Rantanen et al.,


2022). This rapid warming simultaneously affects permafrost, hydrology and surface vegetation, with all of these processes having the potential to alter the generation of alkalinity (Drake et al., 2018). Permafrost thaw causes the Arctic terrestrial freshwater system to increase its connectivity between surface waters and deeper groundwater pathways (Striegl et al., 2005; Walvoord and Striegl, 2007) and may ultimately move from a surface water-dominated to a groundwater-dominated system (Frey and McClelland, 2009). Thawed soils allow longer residence times of infiltrating surface water, thereby facilitating more



contact with unweathered mineral surfaces and additional mixing with mineral-rich groundwater, resulting in higher alkalinity
fluxes (Drake et al., 2018). So far, studies about fast feedback between alkalinity generation from rock weathering and climate
change in northern high latitudes have investigated the impact of a varying glacial cover (Gislason et al., 2009) or retrogressive
thaw slumps (Zolkos et al., 2020) or focused on large Arctic river systems (Drake et al., 2018) and on subcatchments of the
circumboreal (Tank et al., 2012). An in-depth study of the weathering processes in catchments of the silicate-rich
Fennoscandian Shield in northern Norway, potentially influenced by a changing climate, which could lead to an enhanced
alkalinity generation, is still missing.

Our aim in the present study is to identify the weathering pathways responsible for alkalinity generation in a small subarctic
catchment on the mountainside of Iskorasfjellet in northern Norway mainly dominated by silicate minerals and characterized
by sporadic permafrost during early fall of 2020. In particular, we aimed at distinguishing between different weathering agents
(dissolved $CO_2$ vs. other acids) and different source minerals (silicate vs. carbonate) under different environmental forcing,
thereby analysing the potential for $CO_2$ drawdown. In addition to the detailed analysis of the Gaskabohki watershed at
Iskorasfjellet, we extended the investigation of controlling factors on alkalinity generation to several other catchments, some
of which stretch as far as the Barents Sea with varying extent of permafrost, in order to establish broader implications and
evaluate how climate change might provide feedback for $CO_2$ sequestration in this subarctic region.

## 2 Materials and methods

### 2.1 Study site

The Gaskabohki watershed (catchment area = 0.7 km²) is a headwater catchment located on the mountain slope of
Iskorasfjellet, Karasjok Municipality, northern Norway (Fig. 1). Iskorasfjellet is situated inland on the Finnmarksvidda plateau
(300-500 m above mean sea level (amsl)), with local peaks rising above 600 m amsl), which borders Finland to the south and
east. The Finnmarksvidda plateau was completely ice-covered during the Pleistocene glaciations. Due to glacial activity,
ground-moraine, glaciofluvial, and glaciolacustrine sediments were accumulated on the surface geology (Sollid et al., 1973).
Iskorasfjellet was deglaciated ~10,900-10,800 cal. yr BP (Stroeven et al., 2016). The Gaskabohki catchment is underlain by
quartzite and arkose, in places with layers of different shales (NGU, 2022). Therefore, the alkalinity-bearing lithology is
dominated by feldspar, with minor contribution of calcite within the shales. Just before the outlet, however, a small area of
partly calcareous quartz feldspar shale (0.5% areal proportion of the entire catchment area) underlies the catchment. This area
coincides with a wider riparian zone (Fig. 1d). Besides the main channel, drainage gullies are present which were dried out at
the time of sampling in the fall.



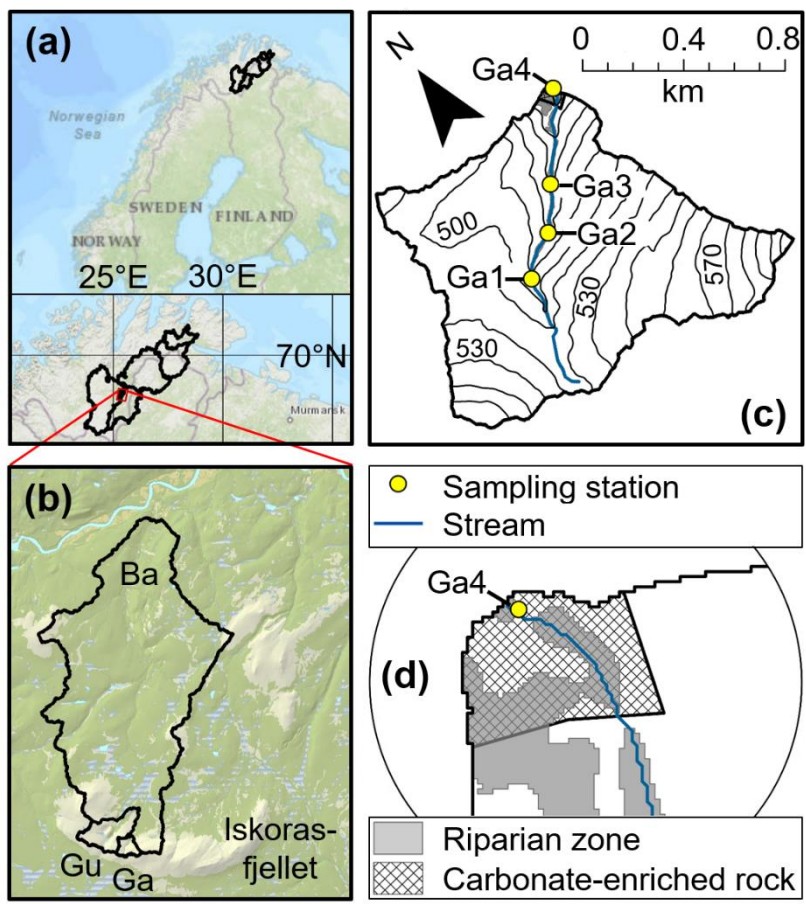

**Figure 1: Study area. (a)** Location of all studied basins in Fennoscandia. Background map from Esri, HERE, Garmin, FAO, NOAA, USGS (ESRI, 2022). **(b)** Zoom on the study area at Iskorasfjellet with the Bahkiljohka (Ba), Guovzilbohki (Gu) and Gaskabohki (Ga) catchments. Background map is the ArcGis web map "Topografisk Norgeskart" (ESRI, 2022). **(c)** Zoom on the Gaskabohki headwater catchment with sampling stations Ga1 – Ga4 (Gaskabohki 1 – Gaskabohki 4). Contour interval = 10 m. **(d)** Zoom on the outlet of the Gaskabohki catchment with the wider riparian zone overlapping with an area of partly calcareous quartz feldspar shale (carbonate-enriched rock).


From the top of the mountain at 644 m amsl., the landscape slopes down (~3% slope) to the Iskoras peat plateau at ~380 m amsl. The peat began to form around 9,800 cal. year B.P. in the form of wet fens, which were prevalent during most of the Holocene. Dry surface conditions associated with permafrost peat plateau aggradation developed around 950–100 cal. year B.P., probably caused by the Little Ice Age cooling (Kjellman et al., 2018).

The Iskoras peat plateau is enclosed by the Bahkiljohka catchment (catchment area = 78 km$^2$), which is underlain by an increased proportion of partly calcareous quartz feldspar shale (up to 34% of areal carbonate extent (NGU, 2022)) when compared to the Gaskabohki catchment (0.5%). From Iskorasfjellet, the sampling area stretches out further northeast, following the larger rivers Karasjohka and Tanaelva, until the Tanafjord. While the Bahkiljohka catchment is characterized by isolated patches of permafrost (catchment average permafrost probability = 0.04), the Gaskabohki watershed as well as the rest of the





studied area show sporadic permafrost (catchment average permafrost probability = 0.10-0.17) (Obu et al., 2019). Tundra
vegetation (e.g. lichen crusts, *Betula* shrubs and *Empetrum nigrum* ssp. *hermaphroditum*) dominates at Iskorasfjellet. Below
an elevation of ~570 m, mountain birch (*Betula pubescens* ssp. *czerepanovii*) forest is also present.

The climate of Finnmarksvidda is continental. For the last six years (September 2014 - August 2020, longest continuous record
at Iskorasfjellet), the mean summer (June-July-August) and winter (December-January-February) air temperatures were 8.7 °C

and -9.3 °C, respectively (measured at the meteorological station at Iskorasfjellet, 591 m amsl, SN97710; (Seklima, 2020)).
Compared to the air temperature normal (1961-1990), temperatures increased by 0.2 °C and 4.2 °C, respectively. Mean
monthly precipitation sums during the summer and winter for the last six years were 66 mm and 38 mm, which correspond to
117% and 237% of the monthly precipitation sums of the climate normal, respectively. The annual precipitation averaged over
the last six years, from September 2017 – August 2020, was 492 mm (measured at the meteorological station in Karasjok,

131 m amsl, SN97251, 20 km from Iskorasfjellet; (Seklima, 2020)). The mean daily snow depth during the snow season
(October – May) was 29 cm (measured at the meteorological station at Iskorasfjellet, 591 m amsl, SN97710; (Seklima, 2020)).
During the sampling campaign in fall 2020, a precipitation sum of 7 mm was measured, which is about half of what was
recorded over the same period averaged over the six years prior to sampling (Seklima, 2020). Thus, the fall of 2020 was
comparatively dry. There was no snowfall and the mean daily air temperature averaged over the sampling period was 3.9 °C,

which is ~2.7 °C higher than during the same fall period 2014-2019 (Seklima, 2020).

## 2.2 Water sampling

From September 22$^{nd}$ to October 6$^{th}$ 2020, we collected water samples daily from the outlet of the Gaskabohki catchment (Ga)
to investigate how DIC and AT may change under different environmental forcing (changing precipitation and temperature).
We further sampled weekly (3x) at three stations upstream of the outlet to track the changes of the carbonate system with

distance from the spring. Besides this temporal examination, we also expanded the investigation spatially, by collecting water
samples from seven further catchments. These catchments included the Guovzilbohki headwater catchment (Gu), which shows
similar topographic properties as the adjacent Gaskabohki watershed, but a higher carbonate-to-silicate bedrock ratio; the
Bahkiljohka catchment (Ba) which is characterized by the lowest permafrost probability; and five larger catchments (catchment
area = 4,900-15,000 km$^2$). In that way, we were able to investigate further controlling factors (catchment area, roughness,

permafrost probability, peatland cover, EVI (mean enhanced vegetation index), and bedrock lithology) on alkalinity
generation.

At all sampling sites, we collected water samples for DIC and AT, $\delta^{13}$C-DIC, major elements, and stable water isotopes ($\delta^{18}$O-
H$_2$O and $\delta^2$H-H$_2$O). For DIC and AT analysis, we collected the river water directly into 300-mL BOD bottles, added 300 μL
of saturated mercury chloride solution and sealed the bottles with ground-glass stoppers, Apiezon type M grease and plastic

caps (no-head space). For $\delta^{13}$C-DIC analysis, the solution was membrane-filtered (0.45 μm pore widths) into 12-mL glass
vials, 10 μL of saturated mercury chloride solution were added, and the vials were sealed with a septum without head space.
For cation measurements, we filtered the surface water (0.45 μm pore widths) into 50 mL acid-washed tubes and acidified the



samples with 50 μL of concentrated trace metal-grade HNO₃. For anion measurements, we filtered (0.45 μm pore widths) the samples into 15 mL tubes, from which we later took an aliquot (~1.5 mL) for stable water isotope analysis. All samples were

stored in the dark at ambient temperature (~1-10 °C). We measured stream temperature and electric conductivity using a precalibrated WTW Multi3430 with IDS TetraCon 925, and turbidity using a precalibrated HACH 2100Qis. Finally, at the outlet of the Gaskabohki catchment, we performed a discharge measurement once a day, at the same time as taking the water samples, by recording stream velocity using a Marsh-McBirney Model 2000 Flo-Mate portable flow meter at increments equal to ~10% of the stream width. By measuring the corresponding stream depth, we were able to calculate stream discharge from

the product of stream velocity and cross-sectional area.

## 2.3 Hydrochemical analyses

We analysed the AT concentration by performing a potentiometric titration using a Metrohm 888 Titrando with an Aquatrode pH probe. The recovery was ≥99%. We used a Marianda VINDTA 3C (Versatile Instrument for the Determination of Titration Alkalinity) to determine the DIC concentration by coulometric titration. The precision was ±2 μmol L⁻¹ (Shadwick et al.,

2011). AT and DIC measuremements were calibrated against certified reference materials (CRMs) provided by Andrew Dickson (Scripps Institution of Oceanography). We calculated pH and pCO₂ by using the program CO2SYS (Pierrot et al., 2011), providing AT, DIC and water temperature, and using the freshwater equilibrium constants from Millero (1979). δ¹³C-DIC were measured by means of continuous-flow isotope-ratio-monitoring mass spectrometry (CF-irmMS) using a Finnigan MAT 253 gas mass spectrometer coupled to a gas bench (GasBench II, Thermo Fisher Scientific) via a ConFlo IV continuous

flow interface following the procedure described by Winde et al. (2014). Precision of δ¹³C-DIC analysis was better than ±0.1‰. Results are given versus the VPDB standard. Stable water isotopes (δ¹⁸O-H₂O and δ²H-H₂O) were measured by means of cavity ring-down spectroscopy (CRDS) using a Picarro L2140-I system. International besides inhouse standards were used to scale the water isotope measurements (Böttcher and Schmiedinger, 2021). Stable isotope results are given in the conventional δ-notation versus the VSMOW standard and had a precision of better than ±0.06‰ for oxygen and ±0.3‰ for

hydrogen isotopes (Böttcher and Schmiedinger, 2021). All stable isotope data given in '‰' are equivalent to 'mUr' (milli Urey; Brand and Coplen (2012)). Multielement composition (Al, Ba, Ca, Fe, K, Mg, Mn, Na, P, Si, Sr) was analysed by inductively coupled plasma optical emission spectrometry (ICP-OES) using a Perkin Elmer Optima 8300DV spectrometer. We determined anion concentrations (Br⁻, Cl⁻, F⁻, NO₃⁻, SO₄²⁻) using a Thermo Fisher Scientific Dionex ICS-2100 ion chromatograph. The precision for multielement and anion analyses was ±10%. The complete hydrochemical data set will be

available through PANGAEA.

## 2.4 Geospatial analyses

We delineated stream networks and watershed areas using the SAGA-GIS modules "Channel Network and Drainage Basins" and "Upslope Area" (Conrad et al., 2015) in QGIS 3.22.2 (QGIS.org, 2022) from the gridded digital elevation model





ArcticDEM (Porter et al., 2018) and used ArcGIS Pro 2.8.0 (ESRI, 2022) to create the maps. While we used the full resolution
model (2 m) for the two small headwater catchments Gaskabohki and Guovzilbohki, we used the 32 m-resolution one for the
larger watersheds.

We calculated the catchments' mean values of terrain roughness, permafrost probability, peatland cover, enhance vegetation
index (EVI) and bedrock lithology. We computed terrain roughness, which is indicative of the potential for physical erosion
(Riley et al., 1999), by using the GDAL tool "Roughness" (Rouault et al., 2022). We determined permafrost probability by
using the Northern Hemisphere permafrost map based on TTOP (temperature at the top of the permafrost) modelling for 2000–
2016 at 1 km² scale, which performs well in sparsely vegetated tundra regions and mountains (Obu et al., 2019), reflecting our
study region. We determined the areal proportion of peatland cover, including mires, from the peatland map of Europe
(Tanneberger et al., 2017). EVI can serve as a proxy for vegetation productivity (Huete et al., 2002). We used the EVI of the
MODIS vegetation index (VI) products, which is based on MODIS data with a resolution of 250 m from October 6th 2020
(Didan et al., 2015). We determined the lithological coverage of the smaller basins (Gaskabohki, Guovzilbohki and
Bahkiljohka) from the Norwegian bedrock map "Berggrunn N250" (NGU, 2022) by calculating the area of the individual rock
types (e.g. quartzite) as a percentage of the total catchment area. For the comparison with the larger catchments, we used the
global lithological map database GLiM, which contains 16 lithological classes (Hartmann and Moosdorf, 2012).

We determined the riparian zone in the Gaskabohki catchment by first calculating the slope per 2 x 2 m DEM cell and then,
starting from the streambed, identifying all cells belonging to the riparian zone that had a slope less than 10° (gradient of
1 : 5.7).

## 2.5 Modelling of soil respiration

We used the streamCO$_2$-DEGAS model by Polsenaere and Abril (2012) to calculate initial soil pCO$_2$. The model first simulates
the decrease in pCO$_2$ and increase in δ$^{13}$C-DIC that occur along the stream watercourse during degassing, starting from an
assumed initial soil pCO$_2$ and ending at the in situ pCO$_2$ or δ$^{13}$C-DIC. Subsequently, soil pCO$_2$ is adjusted until pCO$_2$ and
δ$^{13}$C-DIC simultaneously reach the in situ measured values. Soil organic matter isotopic composition and the isotopic
fractionation of CO$_2$ in the soil due to selective molecular diffusion of the gas through the soil pores are considered. The model
is applicable for small, unproductive streams (assumption of insignificant primary production in the aquatic system) with acidic
pH (4.6 – 7.2) and demands the input variables of AT, δ$^{13}$C-DIC, pCO$_2$, and stream temperature (Polsenaere and Abril, 2012).
We assumed a proportion of non-carbon based acid induced alkalinity generation of 0.2 and a proportion of in-stream
respiration of 0.



## 3 Results and discussion

### 3.1 Weathering and alkalinity generation in the Gaskabohki catchment

During the study period of fall 2020, the Gaskabohki catchment showed a mean AT concentration (± standard deviation) of
125 (± 5) μmol L$^{-1}$ and a mean DIC concentration of 148 (± 21) μmol L$^{-1}$. As the Gaskabohki catchment is mainly underlain
by quartzite and arkose, thus the alkalinity-bearing mineral being feldspar, it is reasonable that the measured AT concentration
exactly matches the concentration that Meybeck (1987) found for catchments draining pure silicate bedrock.

### 3.1.1 Carbonic-acid induced carbonate weathering dominates alkalinity generation

The mean sum of major cation concentrations ($Na^+$, $K^+$, $Ca^{2+}$, $Mg^{2+}$) in the Gaskabohki stream was 120 (± 4) μmol L$^{-1}$. This is
consistent with the lower range found for rivers of the Canadian Shield in the Grenville Province which are characterized by
a similar catchment geology (Millot et al., 2002). The average $Ca^{2+}/Na^+$ and $Mg^{2+}/Na^+$ molar ratios in the Gaskabohki stream
were 0.74 (± 0.05) and 0.49 (± 0.02), respectively. These values are consistent with the ones found by Meybeck (1986) for
quartz sand and sandstone watersheds, and slightly higher than the ones calculated by Gaillardet et al. (1999) for the global
silicate end-member. As we did not correct our data for precipitation due to the lack of rainwater composition data at the study
site, these molar ratios are likely underestimated. Therefore, the weathering load would move closer to the carbonate end-
member. In our data set, the highest $Ca^{2+}/Na^+$ molar ratios (up to 0.83) are associated with both the highest AT and the highest
electrical conductivity values, indicating enhanced carbonate weathering. Considering that our ionic concentrations are not
corrected for rainwater input, these elevated $Ca^{2+}/Na^+$ molar ratios, if corrected, should approach the molar ratio ($Ca^{2+}/Na^+ >$
1) reported by Oliva et al. (2004) for a weathering load that is extensively influenced by trace minerals such as calcite in high
elevation systems draining granitic environments.

The DIC system of the Gaskabohki catchment is neither completely kinetically- (mineral weathering reactions) nor completely
equilibrium- (mixing with atmospheric and biotic $CO_2$) controlled. It is controlled both by $HCO_3^-$ production from carbonate
weathering and by the $CO_2$ pool in the soil (Fig. 2a). While the DIC system mainly consists of $HCO_3^-$ ions (~83 (± 7) %), $CO_2^*$
plays a minor role (~6 (± 9) %) and $CO_3^{2-}$ ions (~1 (± 3) %) are negligible. We found that AT increases linearly with electric
conductivity (adjusted $R^2$ = 0.81), suggesting that the Gaskabohki catchment is mainly controlled by mineral weathering (Fig.
2b). We excluded the measurement with the highest daily precipitation sum of 4.7 mm (4.3 mm within 3 h) from the linear
regression (Fig. 2b, (i)), because this measurement showed a distinctly higher $\delta^{18}$O-$H_2$O value when compared to the other
measurements, thus indicating the highest proportion of surface water. Furthermore, it also showed a distinctly higher turbidity.
As mineral weathering in headwater catchments is generally associated with groundwater inputs (Shin et al., 2011), we thought
this exclusion reasonable. Other authors (Hill and Neal, 1997) explained decreases in AT associated with increases in electric
conductivity with enhanced soil- and surface-derived components, such as nutrients and organic acids. We can also observe
this in our data set, where we detected a nitrate signal in the samples which were taken during a day with rainfall or on up to
four days that followed, before falling under the detection limit, indicating a hysteresis effect (Fig. 3).




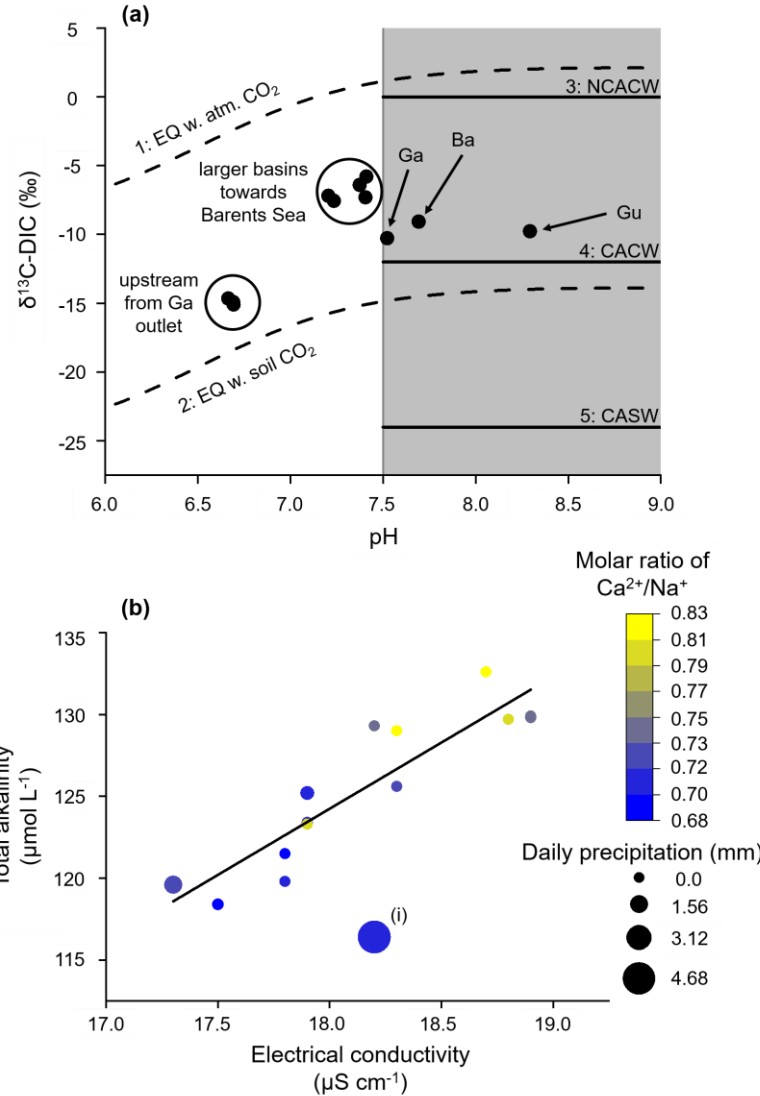

**Figure 2: Equilibrium- vs. – kinetically-controlled systems. (a)** If the pH of the stream water is below 7.5, then the DIC system is equilibrium-controlled, meaning that the stream DIC is in equilibrium with a large pool of either atmospheric (1) or soil (2) $CO_2$. The $\delta^{13}$C-DIC values (dashed lines) were calculated based on the temperature-dependent fractionation factors that partition the carbon isotopes among the DIC species (Zhang et al., 1995). If the pH is above 7.5, the DIC system is kinetically-controlled and three weathering pathways can be identified. Depending on the $\delta^{13}$C-DIC value, the DIC originates from (3) NCACW (non-carbon based acid induced carbonate weathering), (4) CACW (carbonic acid induced carbonate weathering), (5) CASW (carbonic acid induced silicate weathering). Ga: Gaskabohki catchment; Gu: Guovzilbohki catchment; Ba: Bahkiljohka catchment. The figure was adjusted from Lehn et al. (2017). **(b)** AT as a function of electrical conductivity for the Gaskabohki catchment. The linear increase of AT with electrical conductivity indicates that mineral weathering is a major control in this catchment (y-intercept = -21.2 µmol L$^{-1}$, slope coefficient = 8.1). (i) We excluded this data point from the linear regression, as it shows the highest $\delta^{18}$O-H$_2$O value, thus indicating the highest proportion of surface water.





We assumed the $\delta^{13}$C-DIC endmember for carbonic acid induced carbonate weathering to be at about -12‰, as the vegetation at Iskorasfjellet is of C3 type characterized by a $\delta^{13}$C value of about -27‰ (Kjellman et al., 2018). With the diffusive fractionation in low temperature waters, such as during fall in this subarctic region, causing a positive shift of about 3‰, the $\delta^{13}$C value of the $CO_2$ pool would be about -24‰ (Deines et al., 1974; Michaelis et al., 1985; Zhang et al., 1995; Lehn et al.,

2017). When dissolving carbonate minerals with a $\delta^{13}$C value of about 0‰, the resulting $\delta^{13}$C-DIC would thus be about -12‰. If the weathering agent is a non-carbon based acid, the $\delta^{13}$C-DIC endmember for carbonate weathering will be about 0‰. As the mean $\delta^{13}$C-DIC of the Gaskabohki catchment was -10.3 ($\pm$ 1.3) ‰, we attribute the alkalinity generation in the basin entirely to the dissolution of accessory carbonate rock by carbonic acid generated from soil respiration. We suggest that the small deviation from the carbonic acid induced carbonate weathering $\delta^{13}$C-DIC endmember of -12‰ to slightly less negative

values is caused by a small contribution from non-carbon based acid induced carbonate weathering. We believe, that the primary weathering agent of silicate minerals were non-carbon based acids, with no generation of DIC and AT.

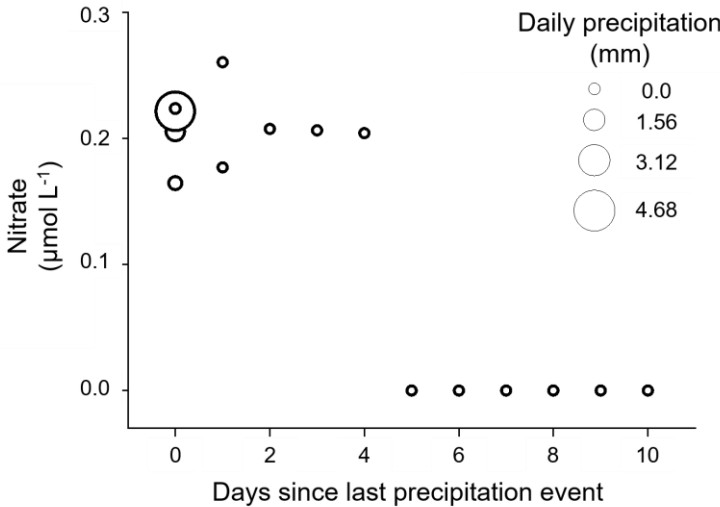

**Figure 3: A nitrate signal is associated with a precipitation event and the days that follow.** On the fifth day after the last precipitation event, we could not detect any more nitrate.


### 3.1.2 Carbonate weathering by sulphuric acid decreases alkalinity concentration

During the study period, AT was positively correlated with the concentration sum of $Ca^{2+}$ and $Mg^{2+}$ (adjusted $R^2$ = 0.57, Fig. 4a), which further confirms that alkalinity is produced from carbonate weathering alone. For one equivalent of a divalent cation, we found about two equivalents of alkalinity in our samples, as would be expected from carbonic acid induced carbonate

weathering (see Eq. (1)). AT concentrations that fall below the regression line are associated with a high molar ratio of $SO_4^{2-}$/AT. We explain this with $Ca^{2+}$ and $Mg^{2+}$ originating from carbonate minerals that were dissolved by sulphuric acid. Thus, no





AT, but $SO_4^{2-}$ was generated. An indicator of the presence of sulphuric acid in the Gaskabohki catchment is that we observed a negative alkalinity concentration at a theoretical electrical conductivity of 0 (according to the linear regression in Fig. 2b, y-intercept = -21.2 μmol L$^{-1}$). We believe that sulphuric acid balances this negative alkalinity in the Gaskabohki catchment.

Sulphuric acid could be generated from the oxidation of pyrite or be a remnant of acid rain. Minor occurrences of pyrite are present in the upper parts of the Palaeoproterozoic bedrock of the Fennoscandian Shield (Sandström and Tullborg, 2009). As our study area is situated on the Karasjok Greenstone Belt, which forms the westernmost unit in a Palaeoproterozoic tectonic belt (Braathen and Davidsen, 2000), the generation of sulphuric acid by the oxidation of pyrite is a reasonable explanation. Even though acid rain deposition has decreased considerably since the end of the last century in northern Norway (Aas et al.,

2021), the recovery from acid rain in deeper soil horizons may be highly delayed (Berger et al., 2016; Marx et al., 2017b). This delayed soil acidification was also observed in southern Sweden, where the B2 horizon did not reach its most acidic conditions until 2013, almost 25 years after the sulphur deposition began to decline (McGivney et al., 2019). Our studied area could be influenced by legacy acid rain distributed area-wide from coal burning or, under the assumption of long-distance transport of pollutants, by the emissions of $SO_2$ from the smelters in Nikel and Zapoljarnij on the Kola Peninsula in Russia

(linear distance between these two sites and Iskorasfjellet: ~200 km). A study in the Tibetian Plateau by Yuanrong et al. (2021) found that the sulphate concentration in glacial and permafrost streams was elevated compared to streams in other landscapes and explained this by condensed storage of acid deposition from long-distance transport during lower temperatures and a release during higher temperatures. Since the main wind directions on the Kola Peninsula in Russia are to the south and to the north (Chekushin et al., 1998) and the Russian smelters are located east of Iskorasfjellet, the sulfuric acid could presumably

originate from pyrite oxidation and legacy acid rain from coal burning.

The $\delta^{13}$C-DIC signal also reflects that two different weathering agents in the Gaskabohki catchment dissolve carbonate minerals. It shows a value of about -12‰ typical of carbonic acid induced carbonate weathering at a low molar ratio of $SO_4^{2-}$/AT (Fig. 4b). As the ratio increases, so does the $\delta^{13}$C-DIC, suggesting an increased proportion of sulphuric acid as the weathering agent of carbonate minerals. We found that a logistic function better fit the data than a linear regression (adjusted

$R^2$ = 0.78 and 0.68, respectively), suggesting a change in weathering pathways. Therefore, regarding the $CO_2$ budget impacted by terrestrial weathering, the catchment can switch from a complete $CO_2$ sink to a partial $CO_2$ source. Since the $\delta^{13}$C-DIC depends on the molar ratio of $SO_4^{2-}$/AT and shows no correlation to discharge (Pearson correlation coefficient r = 0.02, not significant at p < 0.01), indicative of turbulence, we assume that the influence of $CO_2$ outgassing along the stream on the $\delta^{13}$C-DIC signal at the catchment outlet can be ignored.





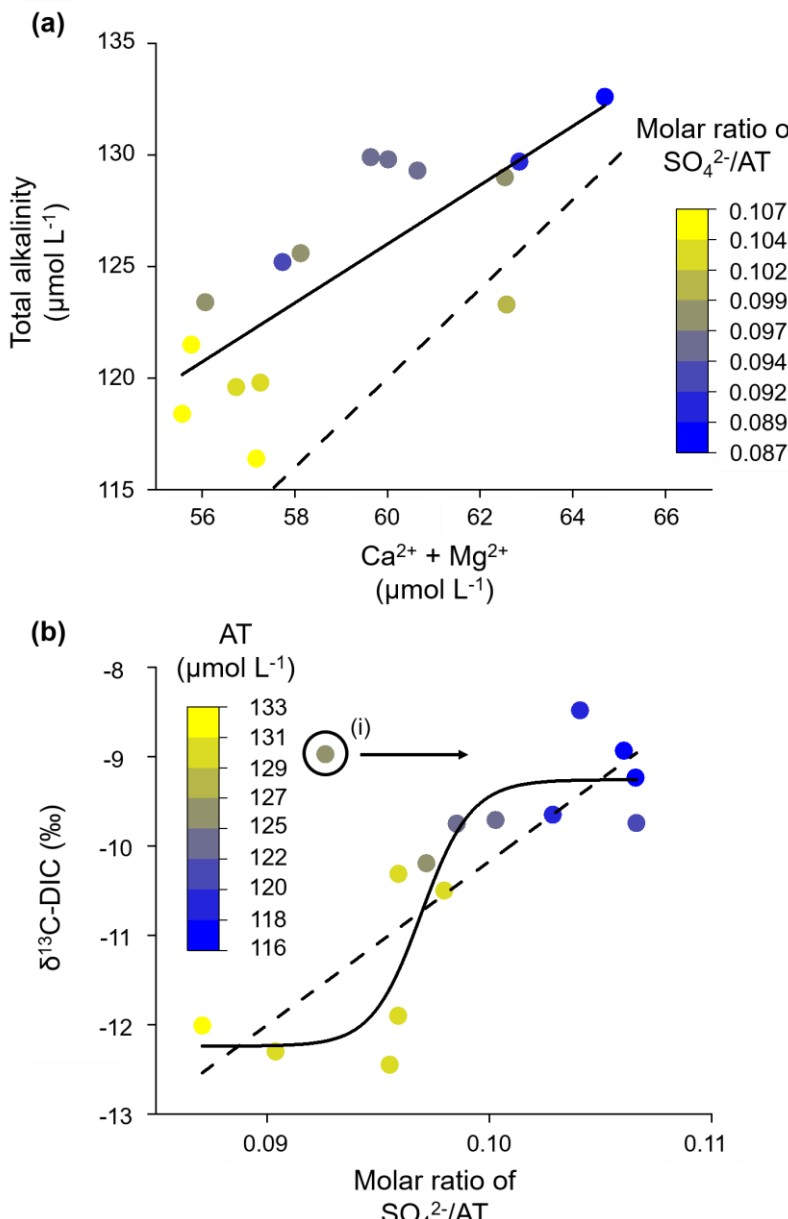


**Figure 4: Alkalinity generation from carbonic acid induced carbonate weathering. (a)** AT increases linearly with the concentration sum of $Ca^{2+}$ and $Mg^{2+}$. At a constant concentration of divalent cations, AT decreases with increasing molar ratio of $SO_4^{2-}/AT$, indicating an increased proportion of carbonate weathering induced by sulphuric acid (release of divalent cations with no generation of alkalinity). Solid line: linear regression (y-intercept = 46.8 µmol $L^{-1}$, slope coefficient = 1.3); dashed line: representing the typical molar ratio of carbonic acid
induced carbonate weathering of AT/($Ca^{2+}$ + $Mg^{2+}$) of 2:1. **(b)** $\delta^{13}C$-DIC increases with the molar ratio of $SO_4^{2-}/AT$. The logistic fit (solid line) yielded a higher adjusted $R^2$ than the linear regression fit (dashed line: y-intercept = -28.5‰, slope coefficient = 183.3), suggesting that a changeover of weathering pathways explains the $\delta^{13}C$-DIC signal better than a linear relationship. (i) We excluded this data point from both model fits, because it has a high pH, likely resulting from in-stream photosynthesis.



We excluded the point with the highest pH (pH = 9.6) from both data fits, which was characterized by a high $\delta^{13}$C-DIC value and a low molar ratio of $SO_4^{2-}$/AT (see (i) in Fig. 4b), as it most likely experienced in-stream photosynthesis which would have lowered the $SO_4^{2-}$/AT molar ratio that was present when initially released into the stream. It is reasonable to assume photosynthesis, because we collected this sample during the afternoon on a day with a long sunshine duration, which would be ideal conditions for photosynthesis (Fig. 5). In addition, the Gaskabohki stream experiences no or only minimal canopy

shading. Therefore, it is likely that in-stream photosynthesis during the course of the day consumed most of the $CO_2$ that was initially produced in the soil and released into the stream, increasing the pH and alkalinity.

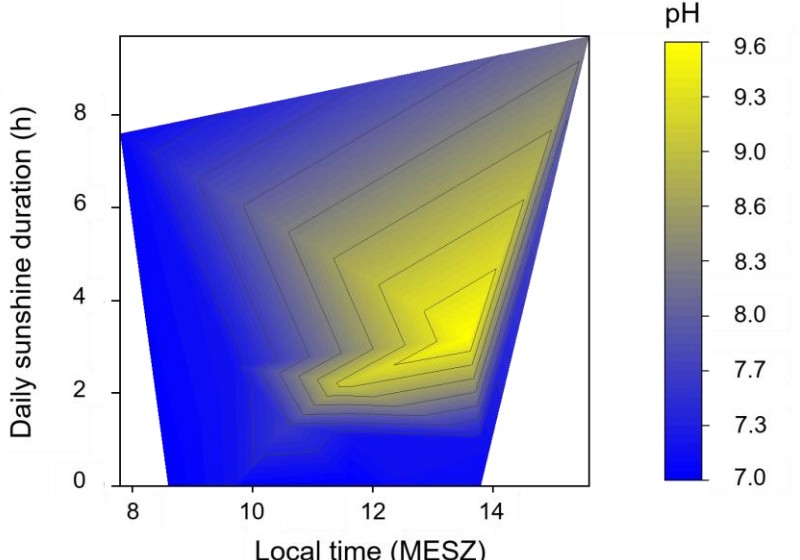

**Figure 5: Heatmap showing the pH for different sampling times and sunshine durations.** A high pH was measured in samples that were taken in the afternoon on a day with a high sunshine duration, indicating in-stream photosynthesis. The pH of samples that were collected in
the morning were not affected by a high sunshine duration.

### 3.1.3 Carbonate-dominated riparian zone controls alkalinity signal

We propose that the AT signal is driven by preferential contribution from two different groundwater sources dominated by different weathering processes: (i) The first source is governed by carbonate minerals being dissolved almost exclusively by
carbonic acid, which results in high AT concentrations and a low $\delta^{13}$C-DIC value of about -12‰. (ii) The second source is governed by an increased proportion of sulphuric acid induced carbonate weathering in which no AT, but $CO_2$ and sulphate are produced and $\delta^{13}$C-DIC is shifted to higher values of about -9‰ (Fig. 4b). Samples that show an intermediate $\delta^{13}$C-DIC signal of about -11 to -9.5‰ represent a mixing of sources. We suggest that the first source (i) is situated in a wider riparian





zone, ~1 km downstream of the spring and thus close to the catchment outlet, representing a hotspot of alkalinity release and

the second source (ii) is stretched out upslope over the remainder of the catchment, representing ~99 % of the entire catchment area.

While almost the entire catchment is underlain by silicates with only accessory calcite from shale as the alkalinity-bearing lithology, the downstream riparian zone coincides with bedrock of partly calcareous quartz feldspar shale. Therefore, the downstream riparian zone bears a greater potential for carbonate weathering. In addition to its advantageous lithological

properties, we propose that the downstream riparian zone, if undisturbed, dominates the alkalinity concentration of the Gaskabohki catchment due to an enhanced hydrological connectivity. Increased soil moisture and shallower water tables near the stream enable both the transport of weathering agent, in the form of soil respired $CO_2$, to the weatherable material and the transport of weathered products from the groundwater to the stream. That the downstream riparian zone is mainly responsible for the alkalinity signal is also reflected in Fig. 2a: While the samples of the three upstream sampling stations, which are

characterized by a narrow riparian zone, are equilibrium-controlled and show a low mean $HCO_3^- / CO_2$ molar ratio of 1.3-1.4, the samples from the catchment outlet, which includes the larger downstream riparian zone, are kinetically controlled and show a higher mean $HCO_3^- / CO_2$ molar ratio of 4.9, indicating enhanced alkalinity generation. Li et al. (2013) also recognized that in the dry season the land cover in the riparian zone much better explained the major elements in the river than the land cover over the entire catchment.

### 3.1.4 Precipitation event temporarily reduces control of the riparian zone

We found that a larger precipitation event on the fourth day of sampling caused a drastic decrease in AT at the outlet of the catchment, reducing AT to 116 µmol $L^{-1}$, the minimum value during our sampling campaign (Fig. 6a, 'day 0' corresponds to the day of the precipitation event). We explain this with a high proportion of surface water from precipitation which diluted the AT signal. This dilution effect can also be observed for the entire data set when plotting AT as a function of $\delta^2H$-$H_2O$: AT

generally decreased with increasing $\delta^2H$-$H_2O$ (Fig. 6b). When surface water mixes with the alkalinity-charged groundwater, the AT concentration measured in the stream is reduced.

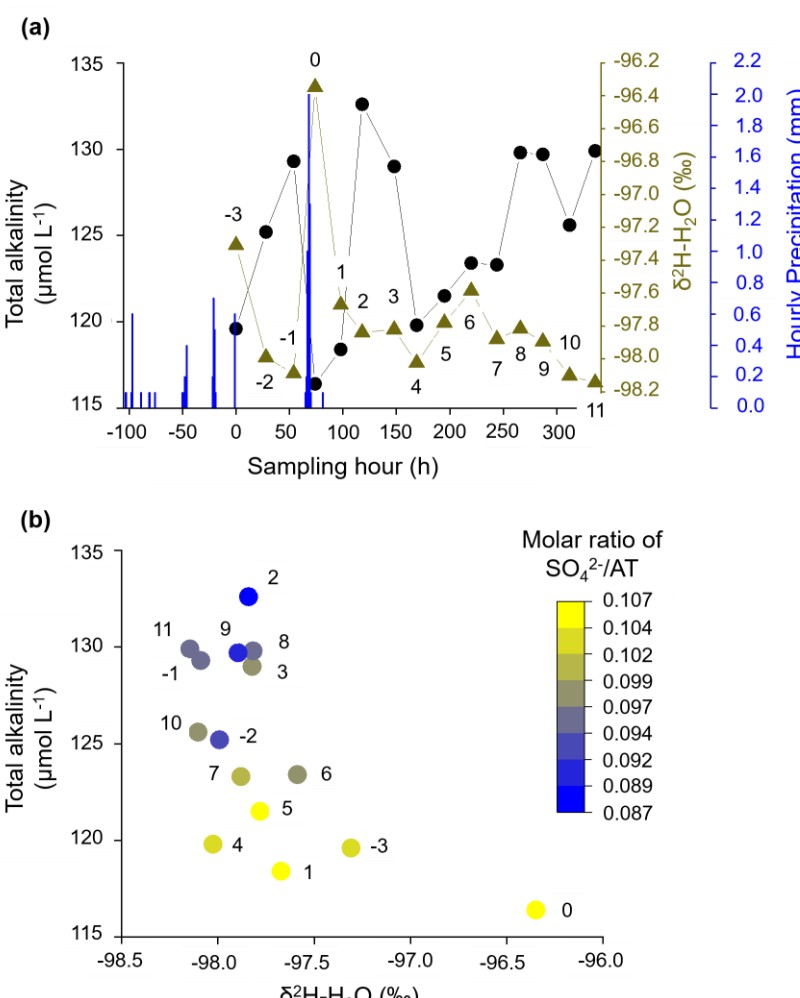

**Figure 6: Alkalinity concentration is influenced by precipitation. (a)** Variation of AT, $\delta^2$H-H$_2$O and hourly precipitation over the course of the sampling campaign. **(b)** AT decreases with increasing $\delta^2$H-H$_2$O, i.e. increasing contribution from surface water. At constant $\delta^2$H-H$_2$O, AT decreases with increasing molar ratio of SO$_4^{2-}$/AT, indicating an increased contribution from the uphill catchment to the weathering load. Numbers next to data points show the number of days before (negative values) and after (positive values) the major rain event (0).

On the second day after the rain event ('day 2'), the AT concentration increased considerably up to 133 µmol L$^{-1}$, the maximum value in our time series. This coincided with a minimum molar ratio of SO$_4^{2-}$/AT, indicating that groundwater from the downstream riparian zone is maximally distributed to the stream water without being diluted by other sources. Petrone et al. (2007) also found that saturated riparian soils and precipitation mainly influence the storm chemistry for a low-permafrost watershed.





After a fast initial surface water response to the precipitation event ('days 0 and 1'), alkalinity-charged groundwater from the downstream riparian zone dominated the stream water signal ('days 2 and 3'). Consequently, we describe the catchment storage-discharge relationship as a clockwise loop with streamflow responding faster than groundwater. On the fourth day after the precipitation event ('day 4'), the AT concentration is reduced once more to 120 µmol L$^{-1}$. However, when compared to the last drop in the AT signal directly after the precipitation event ('day 0'), this decrease cannot be explained with dilution from direct surface water, as the $\delta^2$H-H$_2$O stayed on a low level. We rather attribute this decrease in AT to an increased contribution from uphill groundwater flow, which is dominated by an increased proportion of sulphuric acid-induced carbonate weathering, as the molar ratio of SO$_4^{2-}$/AT is high. The $\delta^2$H-H$_2$O signal slightly increased again ('day 4 – 6'), indicating a delayed contribution from new water of the precipitation event. We suggest that some of the precipitation did not go directly to runoff, but infiltrated the soil of the uphill catchment, mixed with old water and created delayed runoff ('day 5 and 6'). Similarly, in a study about an Arctic watershed in northern Alaska, McNamara et al. (1997) reported that with ongoing thawing season, the storage capacity of the watershed increased, and in conjunction with this, that more new water entered the soil and mixed with old water, as opposed to going directly to runoff. Furthermore, this delayed contribution of uphill groundwater to the stream signal is related to generally larger time scales for subsurface processes. Myrabø (1997) and Camporese et al. (2014) found a similar hysteresis in the catchment storage-discharge relationship. For a riparian zone located in a deeply incised glacial till valley in Indiana, USA, Vidon (2012) reported a quick rise of the water table near the stream and a concomitant decrease in hillslope water contributions to the stream during a storm event. In general, shallower water tables, which characterize riparian zones, frequently respond much more strongly to water infiltration than deeper water tables (Meyboom, 1967). Moreover, for most storms, Vidon (2012) observed the development of a water table down valley gradient. In our study, during the next days after the second decrease in AT ('days 5 – 8'), the AT signal slowly recovered, which coincided with a decrease in the molar ratio of SO$_4^{2-}$/AT, i.e. a declining influence from the uphill catchment (Fig. 6b).

In conclusion, we identify the downstream riparian zone and the hillslope as two key catchment units that collect and transfer water to the stream, as was shown by other authors before (McGlynn and Seibert, 2003; McGlynn and McDonnell, 2003b, 2003a). Usually, in headwater catchments, hillslopes are assumed to be the main contributor to streamflow (Seibert et al., 2009; McGuire and McDonnell, 2010; Vidon, 2012). In our data set, however, we observed that the riparian zone responded more quickly to the precipitation event, probably due to higher antecedent soil moisture and a shallower groundwater table. The stable water isotope composition of all our stream water samples only slightly deviates from the regional and global meteoric water lines (Fig. 7), indicating that the groundwater contribution to stream water most likely originated from shallow groundwater. The average isotopic composition of local precipitation closely reflects the isotopic compositions of shallow groundwater (Fritz et al., 1987). The fall of 2020 was comparatively dry, which may have resulted in particularly pronounced decoupling of uphill groundwater from the stream.





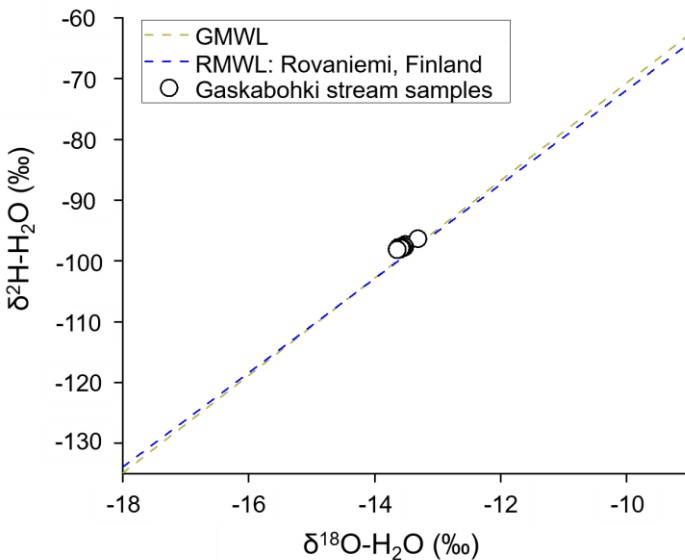

**Figure 7: Stable water isotope composition.** The stable water isotope compositions of the Gaskabohki stream water samples plot closely to the global (GMWL) and regional (RMWL) meteoric water line. RMWL line (y-intercept = 5.6‰, slope coefficient = 7.7) is based on monthly integrated samples collected between January 2004 and December 2019 at the GNIP (Global Network of Isotopes in Precipitation) station in Rovaniemi, Finland (International Atomic Energy Agency, 2020).

On the three days prior to the major precipitation event ('days -3 – -1'), AT showed an increasing trend, while $\delta^2$H-$H_2$O was decreasing. Even though the rainfall intensity on the first day of sampling was not as high as on the fourth day, a contribution of surface water was evident in the $\delta^2$H-$H_2$O signal. Therefore, AT was comparatively low at the beginning of sampling, which is due to a longer, albeit less intense, rain period on a few days prior to the first day of sampling, and thus dilution by surface water. This longer rain period most likely also activated the release of soil-stored nitrate into the stream, which lasted until the
fourth day after the last day with rainfall (Fig. 3). This agrees with observations of Dingman (1971) over a small subarctic catchment with discontinuous permafrost in Alaska, where he found that streamflow recessions are dominated by a combination of tunnel flow under moss-covered parts of the basin and typical groundwater flow through the moss and soils.

### 3.1.5 Silicate weathering with minor alkalinity generation

We suggest that the products from silicate weathering are released together with the products origination from carbonate
weathering by an increased proportion of sulphuric acid from the groundwater of the catchment upstream from the carbonate hotspot riparian zone. Except for this downstream riparian zone, the entire catchment is underlain by silicates with only accessory calcite from shale as the alkalinity-bearing lithology. Based on the $\delta^{13}$C-DIC signal, silicate bedrock is dissolved by sulphuric acid alone, with no generation of DIC. However, some excess alkalinity remains when balancing AT with divalent cations (see dashed line in Fig. 4a, representing the typical molar ratio of carbonic acid induced carbonate weathering of





AT/($Ca^{2+}$ + $Mg^{2+}$) of 2:1). Therefore, we assume that silicate rocks are also weathered by carbonic acid to a small extent. However, when carbonic acid becomes available through soil respiration, it preferentially reacts with the accessory carbonate minerals due to faster dissolution kinetics. Weathering of silicate rocks in the Gaskabohki catchment contributes only minimally to $CO_2$ fixation. Interestingly, theoretical alkalinity production through carbonate weathering (dashed line) deviates most from actual production (solid line) at low concentration sums of $Ca^{2+}$ and $Mg^{2+}$ and high molar ratios of $SO_4^{2-}$/AT. These

are the data points that we associate with increased groundwater contribution from the uphill catchment area, where silicate rock clearly dominates the composition of underlying bedrock over accessory carbonate from shales.

In the previous section (3.1.4), we used the molar ratio of $SO_4^{2-}$/AT as an indicator for the solute contribution from the uphill catchment to the stream water signal at the catchment outlet. This molar ratio is positively correlated with the molar ratio of Na/($Ca^{2+}$ + $Mg^{2+}$) (adjusted $R^2$ = 0.72), indicating that $SO_4^{2-}$/AT can also be used as a tracer for solute contributions from uphill

silicate weathering (Fig. 8a). When the silicate cationic load increases in relation to the carbonate cationic load (Na/($Ca^{2+}$ + $Mg^{2+}$)), AT linearly decreases (adjusted $R^2$ = 0.65, Fig. 8b). As silicate bedrock is weathered by sulphuric acid, $Na^+$ and $SO_4^{2-}$ are the main weathered products from silicate weathering in the groundwater in the uphill catchment, with no generation of AT.





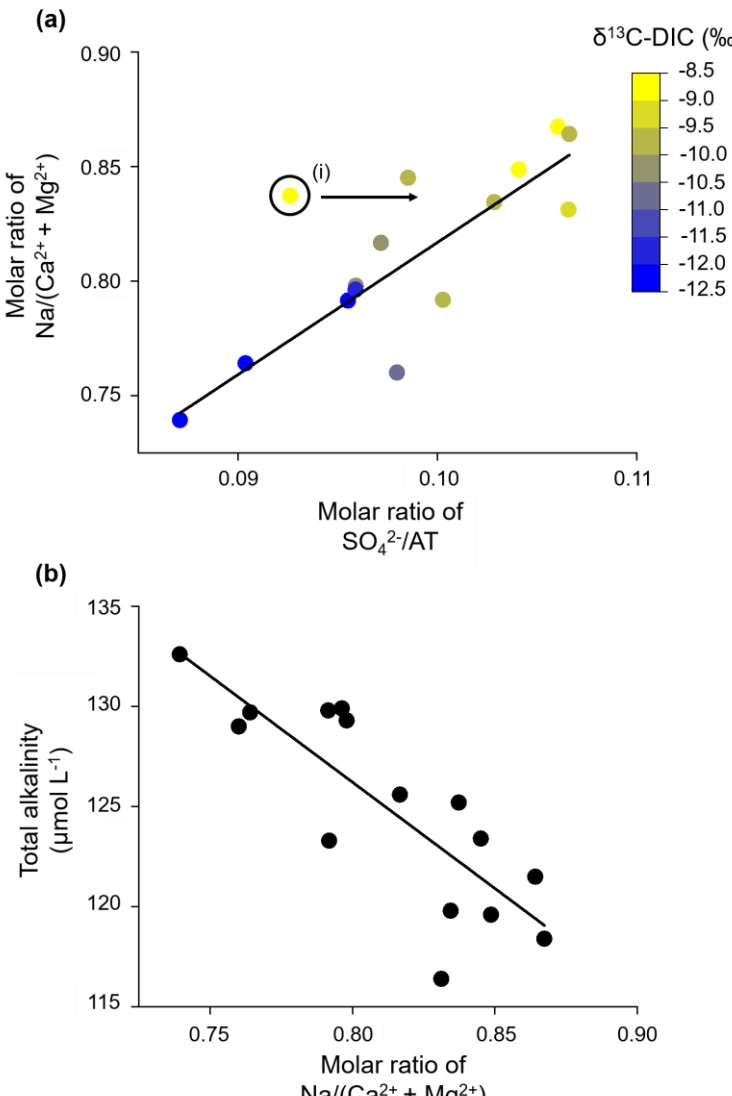

**Figure 8: Uphill silicate weathering. (a)** Molar ratio of $Na/(Ca^{2+} + Mg^{2+})$ increases linearly with the molar ratio of $SO_4^{2-}/AT$, indicating that the uphill silicate cation load behaves similar to the weathering load originating from sulphuric acid induced carbonate weathering. (i) We excluded this data point from the linear regression (y-intercept = 0.24, slope coefficient = 5.8), because it has a high pH, likely resulting from in-stream photosynthesis. **(b)** AT decreases linearly with the molar ratio of $Na/(Ca^{2+} + Mg^{2+})$, indicating that AT is only minimally released from the weathering of the abundant silicate bedrock. Instead, sulphuric acid acts as the main weathering agent; linear regression: y-intercept = 211.1 μmol $L^{-1}$, slope coefficient = -106.1.



### 3.1.4 Availability of weathering agent

The Gaskabohki stream was on average undersaturated with $CO_2$ ($pCO_2$ = 371 ($\pm$ 219) ppm) with respect to atmospheric $pCO_2$, but reached a maximum $pCO_2$ of 651 ppm. By investigating a global compilation of streams and rivers, Marx et al. (2017a)
found that catchments with areas up to 500 $km^2$ show consistently high maximum $pCO_2$ values of ~80,000 ppm. The maximum value we measured during the sampling period in fall 2020 was two orders of magnitude lower. This discrepancy can be explained by the fact that soil respiration in this subarctic region is generally lower compared to the global average, as annual soil respiration decreases with latitude (Warner et al., 2019). Furthermore, with decreased temperatures in the fall, soil respiration was reduced when compared to the summer, during which soil respiration is highly elevated. In a study about $CO_2$
supersaturation in a temperate hardwood-forested catchment, soil $pCO_2$ was modelled between 907 ppm in winter and 35,313 ppm in summer.(Jones and Mulholland, 1998). Another study about soil respiration in permafrost-affected tundra and boreal ecosystems in Alaska and Northwest Canada also recognized the summer time as the main driver of annual soil respiration and calculated that summer months contributed to 58% of the regional soil respiration, winter months contributed to 15%, and the shoulder months contributed to 27% (Watts et al., 2021).
As the mean pH at the "Gaskabohki 1" sampling station, located ~700 m upstream from the outlet and ~400 m downstream of the spring, was 6.7 ($\pm$ 0.0) and thus DIC was present in the form of $CO_2^*$ from respiration of terrestrial organic matter and $HCO_3^-$ from rock weathering, we were able to model the initial $CO_2$ released from soils to surface waters according to the streamCO$_2$-DEGAS model by Polsenaere and Abril (2012). We then used this modelled upstream soil $pCO_2$ as a proxy for the entire Gaskabohki catchment, to which the streamCO$_2$-DEGAS model could not be applied, as the mean pH at the outlet was
7.5 ($\pm$ 0.8), hence most of DIC was only in the form of $HCO_3^-$. With a mean modelled soil $pCO_2$ of 2009 ($\pm$ 471) ppm during fall 2020, the Gaskabohki catchment is characterized by a rather low soil respiration signal when compared to a basin with similar catchment characteristics, but with an annual average temperature of 5.5 °C, situated in northern Czech Republic, which showed a modelled soil $pCO_2$ of 1828-94,454 ppm during fall 2014 (Marx et al., 2018). Our mean modelled soil $pCO_2$ was likely lower due to lower temperatures and less rainfall.

These comparatively low values might explain why the abundant silicate bedrock is almost exclusively weathered by sulphuric acid instead of carbonic acid from soil respiration. When carbonic acid is available, it mainly reacts with the accessory carbonate in the shale layers due to more favourable reaction conditions. We propose that only in the riparian zone along the channel, where soil moisture is highest, soil-respired $CO_2$ is efficiently transported down to the weathering zone.

When looking at individual $pCO_2$ values in the stream at the "Gaskabohki 1" sampling station, we recognized that a
precipitation event caused the highest $pCO_2$ and when no rain fell, a higher temperature resulted in a higher $pCO_2$. We explain this finding with precipitation causing an increased connectivity of $CO_2$-rich soil solutions with the stream, comparable to the increased washout that we observed for soil-stored nitrate (Fig. 3), and higher temperatures causing increased soil respiration. We propose that the in situ weathering rates of carbonate minerals by carbonic acid are moderately activated in the fall due to relatively low $CO_2$ levels. We suggest that these rates are highest during the summer time, when soil respiration is most





activated by elevated temperatures. During that time, groundwater concentrations of alkalinity from carbonic acid induced carbonate weathering should be highest. Therefore, we propose that the riparian zone with its efficient transport of soil $CO_2$ to the weathering zone and its shallow groundwater table is mainly responsible for maintaining high AT concentrations in the Gaskabohki stream.

**3.2 Controlling factors on alkalinity across watershed scales**

No rain fell on the two days on which we extended the sampling to the further catchments up to the Tanafjord. In addition, no precipitation was detected up to seven days before. Thus, a dilution effect, which could be more pronounced in some catchments than others, did not need to be considered. Accordingly, we compared the alkalinity concentration without further normalization and an unconstrained investigation of the factors influencing alkalinity production should be granted.

**3.2.1 Decreasing permafrost probability enhances hydrological connectivity**

First, we will compare the three catchments, which are all situated on the mountainside of Iskorasfjellet: Gaskabohki, Bahkiljohka and Guovzilbohki. Compared to the Gaskabohki catchment, the Bahkiljohka catchment shows a higher stream water pH, therefore plotting further to the right in Fig. 2a, moving more to the site of the end-members for kinetically-controlled mineral weathering reactions. The Bahkiljohka catchment shows a larger catchment area and a lower permafrost probability than the Gaskabohki catchment (Tab. 1). Even more dominated by kinetically-controlled mineral weathering reactions,

however, is the Guovzilbohki catchment. This watershed is a headwater catchment, like the Gaskabohki catchment. In contrast to the Gaskabohki catchment, whose bedrock area is only 0.5% partly calcareous quartz feldspar shale, however, about half (54%) of Guovzilbohki watershed is underlain by this carbonate-enriched bedrock. The Bahkiljohka catchment is characterized by an intermediate proportion (35%) of carbonate-containing lithology. Even though the areal carbonate extent in the Bahkiljohka catchment is not as high as in the Guovzilbohki catchment, it shows the highest AT concentration of 586 μmol L$^{-1}$

(Fig. 9a). We assume that if a certain level of carbonate-containing lithology is present in the catchment, catchment area and permafrost probability control alkalinity generation. This is evident when AT measured in all the catchments studied is plotted as a function of permafrost probability (Fig. 9b). AT linearly decreases with permafrost probability (adjusted $R^2 = 0.28$). Decreases in $HCO_3^-$ flux with increasing permafrost probability was also observed by Tank et al. (2012) for catchment across the circumboreal realm. While permafrost probability, lithology and catchment area are first- and second-order controlling

factor on alkalinity concentration, terrain roughness, EVI and peat cover seem to play a subordinate role in our data set. However, we found that AT generally decreases with increasing terrain roughness and decreasing peat cover, with the Bahkiljohka catchment being an outlier in both correlations. Finally, we recognized a high degree of multicollinearity between permafrost probability and EVI (adjusted $R^2$ when plotting EVI as a function of permafrost probability = 0.63).






**Table 1:** Various catchment properties of the sampled streams and rivers. [1]mt: metamorphics ("wide variety of rocks from shales to gneiss, from amphibolite to quartzite"), [2]vb: basalt-type rocks, [3]pa: plutonic rocks containing quartz, [4]sm: mixed sedimentary rocks ("carbonate is mentioned but not dominant"), [5]sc: carbonate sedimentary rocks (Hartmann and Moosdorf, 2012).

| | Catch-ment area ($km^2$) | Stream length (km) | Rough-ness | Permafrost probability | Areal pro-portion of peatland | EVI | Areal proportion of main lithology classes | | | | |
|---|---|---|---|---|---|---|---|---|---|---|---|
| | | | | | | | mt[1] | vb[2] | pa[3] | sm[4] | sc[5] |
| Gaskabohki 1 | 0.2 | 0.4 | 25.5 | 0.13 | 0.00 | 0.29 | 1.00 | 0.00 | 0.00 | 0.00 | 0.0 |
| Gaskabohki 2 | 0.2 | 0.6 | 25.8 | 0.13 | 0.00 | 0.29 | 1.00 | 0.00 | 0.00 | 0.00 | 0.0 |
| Gaskabohki 3 | 0.4 | 0.8 | 27.8 | 0.13 | 0.00 | 0.28 | 1.00 | 0.00 | 0.00 | 0.00 | 0.0 |
| Gaskabohki 4 (outlet) | 0.7 | 1.1 | 27.7 | 0.13 | 0.00 | 0.28 | 1.00 | 0.00 | 0.00 | 0.00 | 0.0 |
| Guovzilbohki | 2.8 | 3.6 | 25.3 | 0.10 | 0.00 | 0.30 | 1.00 | 0.00 | 0.00 | 0.00 | 0.0 |
| Bahkiljohka | 78 | 20 | 19.8 | 0.04 | 0.06 | 0.30 | 0.76 | 0.16 | 0.00 | 0.06 | 0.0 |
| Karasjohka 1 | 4909 | 162 | 11.5 | 0.17 | 0.16 | 0.27 | 0.69 | 0.10 | 0.08 | 0.11 | 0.0 |
| Karasjohka 2 | 7255 | 187 | 12.9 | 0.14 | 0.18 | 0.28 | 0.67 | 0.16 | 0.06 | 0.08 | 0.0 |
| Karasjohka 3 | 11614 | 297 | 15.9 | 0.14 | 0.15 | 0.27 | 0.76 | 0.11 | 0.04 | 0.06 | 0.4 |
| Tanaelva 1 | 14085 | 357 | 15.8 | 0.13 | 0.15 | 0.27 | 0.69 | 0.11 | 0.10 | 0.07 | 0.5 |
| Tanaelva 2 | 15156 | 384 | 15.9 | 0.13 | 0.14 | 0.28 | 0.65 | 0.10 | 0.10 | 0.12 | 0.3 |

As discussed in section 3.1 in detail for the Gaskabohki catchment, it seems that alkalinity generation is limited by the contact of weathering agent ($CO_2$ from soil respiration) with weatherable material as well as the transport of the weathered products out of the weathering zone into the stream. In the Gaskabohki catchment, the saturated riparian zone facilitates this hydrological transport. In the Bahkiljohka catchment, which shows the highest AT concentration, the low permafrost probability is most likely responsible for enhanced hydrological connectivity. In another study about the effects of permafrost loss on discharge

from a wetland-dominated, discontinuous permafrost basin, Stone et al. (2019) reported that total annual discharge from the channel fen decreased by 2.5 % for every 10 % decrease in permafrost area due to increased surface storage capacity, reduced run-off efficiency, and increased landscape evapotranspiration.



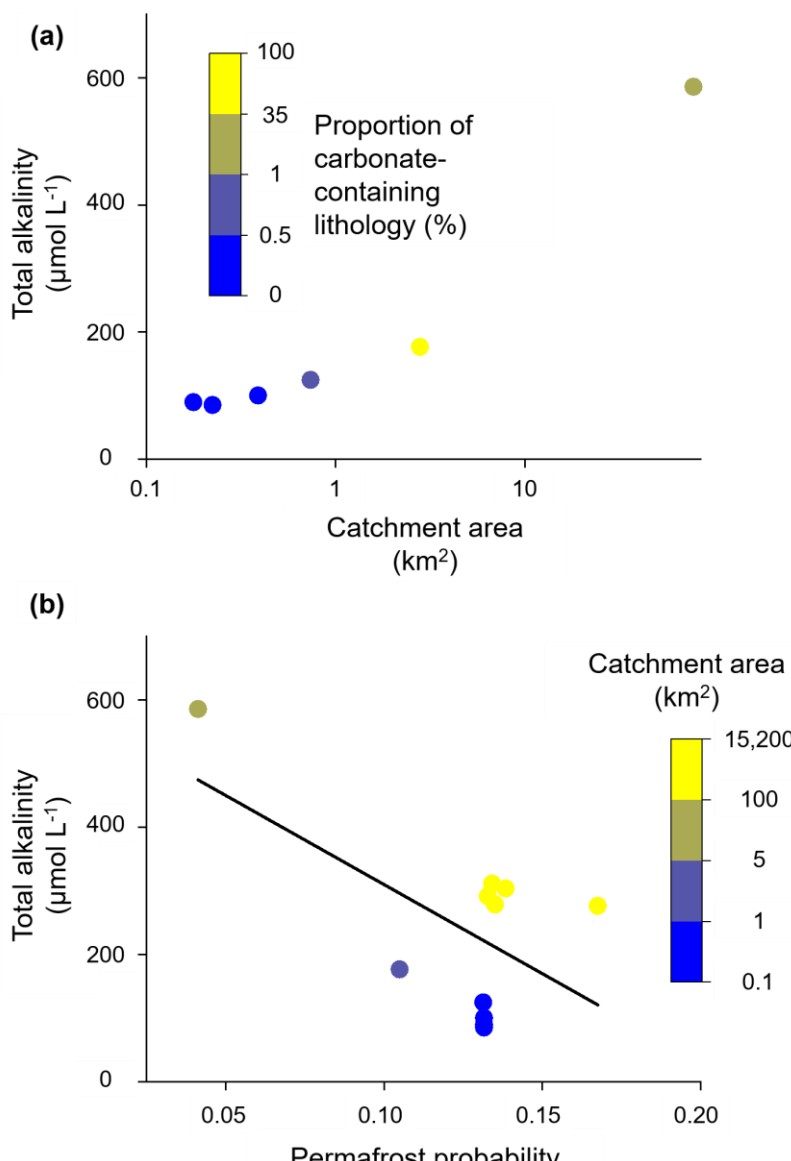

**Figure 9: Controlling factors on alkalinity concentration across watershed scales. (a)** At Iskorasfjellet, AT increases with catchment area and the proportion of carbonate-containing lithology. **(b)** At a larger spatial scale, AT is controlled by permafrost probability, with a low permafrost probability being associated with a high AT concentration. At a similar permafrost probability, a larger catchment area yields a higher AT concentration; linear regression: y-intercept = 590.0 μmol L$^{-1}$, slope coefficient = -2800.6.

At constant permafrost probability and thus similar hydrological conditions, AT increases with catchment area. From the in-depth study of the Gaskabohki catchment, we found that a precipitation event resulted in the highest turbidity values which we





associate with increased sediment supply to the stream. Therefore, intense precipitation events have the potential to activate the fluvial transport of mineral substrate downstream. This new material together with the surface water most likely leave the headwaters quickly. In the larger rivers downstream, however, this material can undergo several weathering cycles, increasing alkalinity. From decade-long hydrometeorological and biogeochemical observations of catchments in the High Arctic, Beel et

al. (2021) deduced that increased late summer rainfall enhanced terrestrial-aquatic connectivity for dissolved and particulate material fluxes. Zolkos et al. (2020) showed that this particulate material, in their study on retrogressive thaw slumps, can rapidly weather during fluvial transport within runoff.

The DIC systems of the three small subcatchments situated within the Gaskabohki headwater catchment and the five larger catchments that drain into the Barents Sea are equilibrium-controlled (Fig. 2a). However, while the DIC species of the three

small catchments mix with biotic $CO_2$, the DIC species of the five larger basins are rather in exchange with atmospheric $CO_2$. For undisturbed headwater catchments on the Peel Plateau in Canada, Zolkos et al. (2020) also reported that stream chemistry reflected $CO_2$ from soil respiration processes.

### 3.2.2 Alkalinity generation from carbonate weathering – from Iskorasfjellet to the Tanafjord

As we found in section 3.1.2 for the Gaskabohki catchment on the temporal scale, AT also linearly increases on the spatial

scale with the concentration sum of $Ca^{2+}$ and $Mg^{2+}$ (adjusted $R^2$ = 0.98, Fig. 10a). Therefore, we propose that alkalinity is produced from carbonate weathering by carbonic acid in all basins that we studied – from Iskorasfjellet to the Tanafjord. While the Gaskabohki and Guovzilbohki headwater catchments with the two smallest concentration sums of $Ca^{2+}$ and $Mg^{2+}$ show the characteristic molar ratio of AT/($Ca^{2+}$ + $Mg^{2+}$) of 2:1, indicative for carbonate weathering by carbonic acid (see dashed line in Fig. 10a), the larger catchments show a reduced molar ratio. This decrease coincides with an increase in the molar ratio of

$SO_4^{2-}$/AT, suggesting that the carbonate minerals are increasingly dissolved by sulphuric acid. In addition to the Gaskabohki catchment, sulphuric acid appears to be present in the other basins, because at a theoretical electrical conductivity of 0, the linear regression of AT as a function of electrical conductivity suggests a negative alkalinity (y-intercept = -16.9 μmol L$^{-1}$, adjusted $R^2$ = 0.99, Fig. 10b). Therefore, we assume that some of the divalent cations originate from sulphuric acid induced carbonate weathering, with no generation of AT, but $CO_2$ and $SO_4^{2-}$. Thus, carbonate weathering in the studied region does

not only consume $CO_2$, but also releases $CO_2$, albeit to a lesser extent. Carbonic acid induced silicate weathering does not appear to contribute to alkalinity generation and thus to $CO_2$ sequestration (Fig. 2a), although silicate bedrock is generally more abundant in the study area than carbonate bedrock according to lithological maps (Tab. 1).


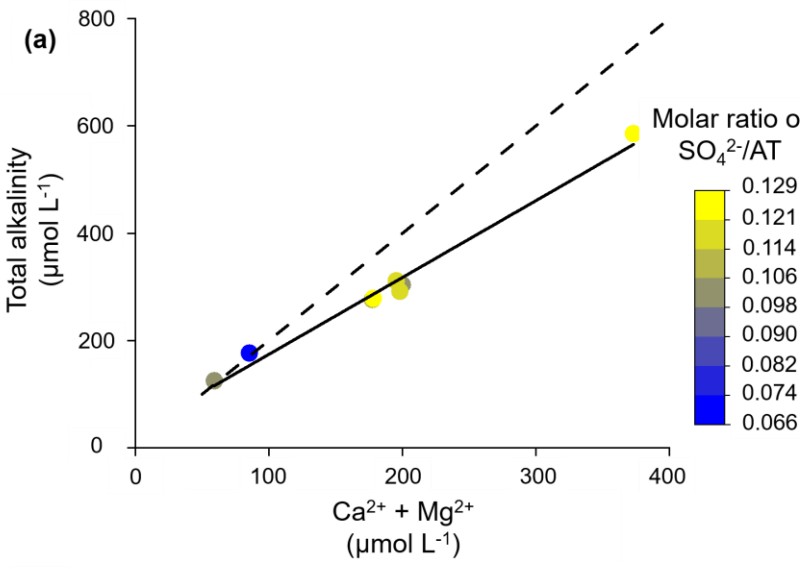

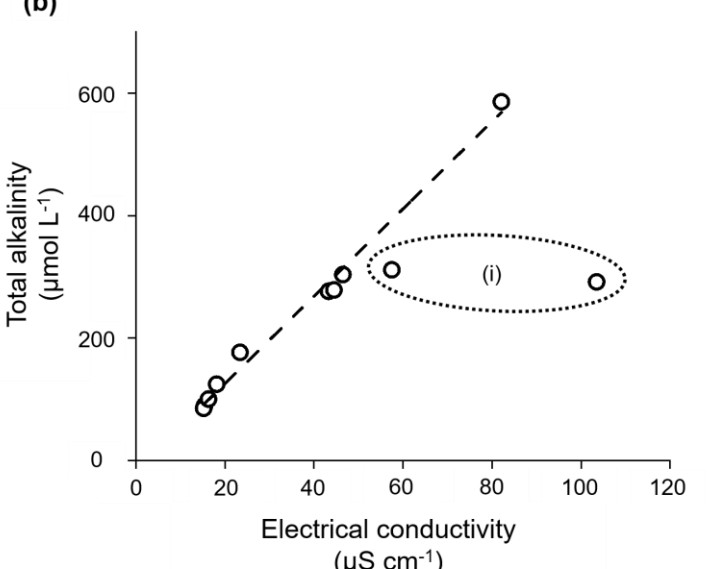

**Figure 10: Alkalinity generation from carbonate weathering across watershed scales. (a)** AT increases linearly with the concentration sum of $Ca^{2+}$ and $Mg^{2+}$. Solid line: linear regression (y-intercept = 31.1 µmol $L^{-1}$, slope coefficient = 1.4); dashed line: representing the typical molar ratio of carbonic acid induced carbonate weathering of AT/($Ca^{2+}$ + $Mg^{2+}$) of 2:1. **(b)** AT increases linearly with electrical conductivity. (i) Dotted ellipse: We excluded these two data points from the linear regression (dashed line: y-intercept = -19.9 µmol $L^{-1}$, slope coefficient = 7.1), as they are characterized by distinctly higher $Na^+$ and $Cl^-$ concentrations (*PANGAEA-DOI-Link*), indicating an influence from the salty fjord and seawater close by.



## 4 Conclusions

In the present study is shown that weathering of accessory carbonate dominates the alkalinity generation in a mainly silicate-dominated subarctic headwater catchment in Northern Norway. The vast Fennoscandian Shield is generally underlain by silicates as the alkalinity-bearing rock type. An area of ~20,000 km$^2$ of the Fennoscandian Shield, however, is classified as
metamorphic rocks with minor carbonate occurrences (Hartmann and Moosdorf, 2012), the same geology that dominates the Gaskabohki catchment. When transferring our results about the Gaskabohki catchment to the Fennoscandian Shield, we propose that alkalinity dynamics in this area are also drastically influenced by carbonate minerals, even though their occurrence may be low. Further, we have found that alkalinity generation by carbonate weathering greatly decreases when the proportion of sulphuric acid as the weathering agent increases. This was particularly evident in the larger catchments towards the
Tanafjord.

For the Gaskabohki headwater catchment, we identified the downstream riparian zone characterized by a carbonate-enriched lithology, as the main contributor to alkalinity in the stream. This riparian zone appears to facilitate both the transport of soil-respired $CO_2$ to the weathering zone und the transport of weathered products to the channel. When undisturbed, high alkalinity concentrations (~130 $\mu$mol L$^{-1}$) are maintained by the inflow of alkalinity-charged groundwater from the downstream riparian
zone. However, after a precipitation event, the alkalinity concentration in the stream was reduced twice due to dilution: 1) immediately after the rain event, alkalinity was diluted by new surface water, and 2) after 4 to 7 days after the rain event, the signal was diluted again by a delayed contribution of uphill groundwater to the stream that contained reduced amounts of alkalinity.

Weathering of silicate rocks by carbonic acid, and hence long-term $CO_2$ sequestration, seems to be limited by insufficient
contact between the weathering agent ($CO_2^*$) and the mineral surface. We expect silicate weathering rates to increase in the Gaskabohki headwater catchment due to climate change in the future, as both the permafrost extent is declining and the annual precipitation is increasing. These two trends will increase the water storage capacity, thereby increasing the contact time of weathering agent and weatherable material. McNamara et al. (1997) suggested that as the thickness of the active layer in permafrost increases, not only will the hydrologic response of streams to precipitation inputs be attenuated due to greater water
storage capacity, but also root and heterotrophic respiration will increase as it occurs almost exclusively in this thawed layer. Therefore, we expect the availability of weathering agent in the form of carbonic acid to increase.

In conclusion, we expect an increase in alkalinity generation from carbonic acid induced silicate weathering in the Gaskabohki catchment in the future, although we believe the feedback to be slow, as silicate weathering rates are several orders of magnitude slower than carbonate weathering rates. We also propose that carbonic acid induced carbonate weathering is likely
to increase. We suppose that alkalinity generation from carbonate weathering is responding faster to a changing climate. We believe that parts of the Fennoscandian Shield will experience a rapid weathering response of accessory carbonate minerals to climate change in the future, resulting in elevated alkalinity levels. Future long-term studies that include more seasons and groundwater sampling are required to confirm this hypothesis.





*Data availability*. All data used in this study is available at PANGAEA (*PANGAEA-DOI-Link*).


*Author contributions*. NL, HL and HT designed the study. NL led the field research, coordinated laboratory analyses, and wrote the first draft of the manuscript. NL measured DIC concentration. AE measured cation and anion concentrations. MEB and JH contributed (isotope) analytical measurements and data interpretation. All authors contributed to manuscript writing and editing.


*Competing interests*. The authors declare that the research was conducted in the absence of any commercial or financial relationships that could be construed as a potential conflict of interest.

*Acknowledgements*. We thank the German Federal Ministry of Education and Research (BMBF) for providing the funding for
this project. NL and HT were supported by "The Ocean's Alkalinity: Connecting geological and metabolic processes and time-scales", BMBF under "Make our Planet Great Again – German Research Initiative", grant number 57429828, implemented by the German Academic Exchange Service (DAAD). We thank M. Treblin and L. Detjen for assistance in the field. JH wishes to thank P. Bartsch for measuring AT concentration.





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
