# Peer review of "Alkalinity generation from carbonate weathering in a silicatedominated headwater catchment at Iskorasfjellet, northern Norway"

_Biogeosciences, 2022_

## Referee Comment (RC2)

[referee-annotated manuscript omitted]

---

## Author Response (AR1)

**Authors' Responses to Reviewers' Comments**

**Reviewer #1**

| # | Reviewer's Comment | Authors' Reponse | Position in Manuscript |
|---|---|---|---|
| 1 | Sulphur is not only derived from pyrite oxidation as stated here, but also from oxidation of stores of reduced sulphur in wetlands (including permafrost), accumulated during anoxic breakdown of organic matter (accumulated through anaerobic microbial metabolic processes). During dry periods or permafrost melting, reduced sulphur is oxidized to $SO_4^{2-}$ and then is typically released in the next flushing event to drainage waters. | Yes, good point. We added the possibility of sulphur being derived from oxidation of stores of reduced sulphur in wetlands (including permafrost), as the reviewer suggested. | L32, L73, L340 |
| 2 | Acid rain is more precisely referred to as "acid deposition" | We changed acid rain to acid deposition. | L74, L345, L348 |
| 3 | Similarly, $NO_3^-$, which can form nitric acid, can also be released through nitrification of reduced nitrogen stores in soils, and not only through fertilizer application or acid deposition. | Also a good point. We added the possibility of nitrate being derived from nitrification of reduced nitrogen stores in wetland soils. | L75 |
| 4 | "decreasing permafrost probability" is not clear. Does it mean likelihood of permafrost loss? | We added a definition of 'permafrost probability' according to Obu et al. (2019) to the methods section: "Permafrost probability is defined as the fraction of ensemble members ($N$ = 200 for each 1 $km^2$ grid cell) with a mean annual ground temperature of 0°C or lower." | L36, L180 |
| 5 | Not clear that this sentence includes DIC generated from heterotrophic respiration, so consider: "biogenic DIC originates from autotrophic respiration, heterotrophic respiration and organic matter mineralization (e.g., photooxidation of DOC)". | We edited the sentence and included heterotrophic respiration. | L86 |
| 6 | Fix punctuation: a clause before a semicolon needs to be independent. | Done! | L120 |
| 7 | Awkward sentence: "Especially carbonate weathering was found to be very responsive to contemporary environmental changes." | We changed the sentence to: "Especially carbonate weathering was found to strongly respond to contemporary environmental changes." | L121 |

| 8 | For the sentence: "Besides the tropical region, northern high latitudes are expected in the future to experience enhanced carbonate weathering and thus a higher carbon-sink function due to increased soil $CO_2$ partial pressures and temperatures". Clarify the driver of increased soil $pCO_2$: do you mean that it is caused by increased atmospheric $CO_2$ or from another source? | We added the sentence "The drivers of the increased soil pCO2 are increased ecosystem productivity and soil respiration (Zeng et al., 2022)." to clarify this. | L123 |
|---|---|---|---|
| 9 | Rapid warming also can affect soil carbon stores. | Okay, we added this. | L130 |
| 10 | Information on the soil types would be helpful. | Good point, we added information on the soil types being present on the Finnmarksvidda plateau. | L158 |
| 11 | Verb tense inconsistent | Okay, we edited this. | L200 |
| 12 | Semi-colons should not be used here as clauses are not independent. | Okay, we edited this. | L202-203 |
| 13 | Should be "enhanced vegetation" | Okay, we edited this. | L247 |
| 14 | A unique method for defining riparian zone – is there a reference for this method? | Other authors (Camporese et al., 2014; Jencso et al., 2009) determined the riparian zones by computing the flow accumulation for the catchment (using a triangular multiple flow-direction algorithm), allowing for the identification of the cells belonging to the stream. Then they identified the riparian zones as those flow accumulation cells characterized by a difference in elevation lower than 3 m compared to the stream cell they drained into. We wanted to keep it a more simple process (not applying a flow accumulation algorithm), but still take into account the special characteristics of a hillslope catchment, so we included slope as a criterion instead of just using a constant distance from the main channel. Camporese et al. (2014) found that the primary control on the nonlinear catchment response to a rainfall-runoff event is exerted by topography, thus we included slope as a criterion. We added a sentence at the end of this paragraph referencing to the findings of Camporese et al. (2014) to justify our decision to use slope as the main criterion for determining the riparian zone. | L260 |
| 15 | How realistic are these assumptions? For discussion - how might alternative values affect the results? | We added a sentence in the methods section that we explored the effects of alternative values on the results. We discussed the | L270, L524 |

| | | | |
|---|---|---|---|
| | | results of this sensitivity analysis in the discussion section '3.1.4 Availability of weathering agent'. | |
| 16 | Spelling error: should be "reasonable" | Okay, we edited this. | L276 |
| 17 | For "Furthermore, it also showed a distinctly higher turbidity." Clarify what "it" refers to. | Okay, we changed "it" to "this measurement". | L298 |
| 18 | The reference weathering processes in riparian zones is not fully correct. Considering that riparian zones are typically defined as linear vegetation zones (often 20 m on either side of watercourses), underlying bedrock doesn't usually follow the riparian zone exactly. Also, please clarify which disturbance of the riparian zone would change the weathering processes or hydrological connectivity. And instead stating that the riparian zone is "responsible for the alkalinity signal", it would be more accurate to state that weathering processes in the carbonate rock located in the lower reaches of the catchment is responsible for this alkalinity signal (L 404). | Yes, we see the potential for confusion. We clarified that the weathering processes in the carbonate rock located in the downstream area of the catchment are responsible for the alkalinity signal, not the riparian zone. We also added "if not disturbed by a precipitation event", clarifying that when the riparian zone is disturbed by a precipitation event (dilution by surface water), the alkalinity concentration in the stream is reduced. | L398 |
| 19 | The description of hydrologic pathways here isn't fully correct and needs fixing to support conclusions, as described below in more detail. In general, to fix this issue, I would suggest replacing "riparian zone" with more precise descriptors of hydrologic pathways.
• E.g., RE: "Increased soil moisture and shallower water tables near the stream enable both the transport of weathering agent, in the form of soil respired $CO_2$, to the weatherable material and the transport of weathered products from the groundwater to the stream." The water table doesn't enable the transport of the weathering agent; rather, weathering agent transport is a function of the hydrological flowpaths in the area.
• E.g., the statement that riparian zones do not respond more strongly to water infiltration than deeper water tables doesn't make sense hydrologically.
• The description of riparian zones collecting water for the stream isn't fully correct, as they are typically discharge zones rather than recharge zones. | Okay, we edited the text accordingly. We said that the riparian zones do respond more strongly to water infiltration. And yes, riparian zones are discharge zones. We edited this. | L402, L443-452 |

| | | |
|---|---|---|
| | • The conclusions in L449, L536, and L631 should be revisited after correcting the hydrological concepts. | |
| 20 | Tunnel flow is more commonly referred to as throughflow or piping. | Okay, we changed it to throughflow. | L470 |
| 21 | Should be " products' " on second instance | Okay, we edited this. | L474 |
| 22 | Statement needs to be clarified and supported better. | Okay, we edited this section. | L475 |
| 23 | Need to explain why the dilution effect would be more pronounced in some catchments than others. Or consider deleting. | We deleted "which could be more pronounced in some catchments than others". | L541 |
| 24 | Could the higher AT concentration be due to lower presence of strong acids, and therefore a result of a higher percentage of weathering being derived from $CO_2$*? | We think this is not the case, as this catchment (Bahkiljohka) shows a reduced molar ratio of AT/($Ca^{2+}$ + $Mg^{2+}$) and an increased molar ratio of $SO_4^{2-}$/AT when compared to the two headwater catchments, suggesting that the carbonate minerals are increasingly dissolved by sulphuric acid (see Fig. 10a). | L554 |
| 25 | Should clarify suspended sediment supply, if based on observations of turbidity. "Substrate" often refers to bed material instead of suspended material, so I suggest removing that term for clarity. | Okay, good point! We will use suspended sediment supply instead of mineral substrate. | L586 |
| 26 | It is more accurate to state that groundwater flows to the stream through the hyporheic zone, not the riparian zone. | Okay, we changed this. | L634 |

**Reviewer #2**

| # | Reviewer's Comment | Authors' Reponse | Position in Manuscript |
|---|---|---|---|
| 1 | What about the role of organic acids in carbonate weathering in the riparian region? | We added a sentence in L111 and a sub clause in L477 concerning weathering by organic acids (see below). We also added a sentence in L339 concerning the weathering of carbonate minerals in particular: "In addition to sulphuric acid, organic acids could act as weathering agents for the carbonate minerals."
We believe, however, that organic acid induced carbonate weathering in the riparian zone plays a minor role, as the $\delta^{13}$C-DIC signal (~-12‰) points to carbonic acid induced carbonate weathering. | L39 |
| 2 | Sentence not clear, kindly rephrase. | Okay, we edited this section. | L42 |
| 3 | thought to be in equilibrium | Okay, we edited this. | L45 |
| 4 | Line number 46 is not in continuation with line 44-45. Kindly check and rephrase. | We added a sentence between both parts to create a better transition. | L46 |
| 5 | This notation can be changed. | Yes, we changed "he" to "Hartmann (2009)". | L53 |
| 6 | Pls rephrase. | Okay, we edited this. | L54 |
| 7 | Rephrase, not clear. | We changed the sentence to: "Especially silicate-dominated regions that are physically active, e.g., during glaciation and tectonism, or that exhibit early stages of weathering due to freshly deposited material or recent deglaciation or uplift, show high weathering loads of accessory carbonate minerals such as calcite, aragonite and dolomite (Jacobson et al., 2002; Jacobson et al., 2003; White et al., 1999; Oliver et al., 2003; White et al., 2005; Moore et al., 2013; Jacobson et al., 2015)." | L55 |
| 8 | Pls list some of the minerals. | Done! We added "calcite, aragonite and dolomite". | L56 |
| 9 | Rephrase. | Done! | L58 |
| 10 | What about pH? This is the most important factor. How alkalinity is a controlling factor when DIC itself is a component of alkalinity? | Under environmental conditions DIC and AT together regulate pH and other parameters of the carbonate system. This can be exemplified by comparing the concentration ranges of DIC and | L62 |

| | | | |
|---|---|---|---|
| | | AT being both in the millimolar range but H+ in the nanomolar range. Furthermore, DIC is not a component of AT, both share the species $HCO_3^-$ and $CO_3^{2-}$. In addition, DIC comprises $CO_2$*, AT also non-carbonate species such as borate and others. | |
| 11 | Pls list other components also. | We added hydroxide ions to bicarbonate and carbonate ions. | L63 |
| 12 | It is not clear, how increment in DIC will not contribute alkalinity? Please see the reference Appelo, C.A.J., Verweij, E., Schäfer, H., 1998. A hydrogeochemical transport model for an oxidation experiment with pyrite/calcite/exchangers/organic matter containing sand. Applied geochemistry 13, 257-268. $HCO_3^-$ may be produced at low $H^+$ concentrations that is in excess $CaCO_3$ condition. | Good point, we clarified this by adding the information about the amount of $CaCO_3$ being present, changing the sentence to: "Assuming that the weathering agent is not carbonic acid but other inorganic acids such as sulfuric and nitric acids, and carbonate is not present in excess but as accessory carbonates, carbonate weathering (shown here for calcite weathering with sulfuric acid) releases $CO_2$, increasing the DIC concentration but not contributing to alkalinity formation." We also added the reference the reviewer suggested. | L70, L72 |
| 13 | In equation 3, for carbonate weathering $CaAl_2Si_2O_8$ is mentioned instead of $CaCO_3$. Pls check. | We wanted to show an example for silicate weathering in equation 3 and chose anorthite. We think that the second part of the sentence in L76-78 makes this clear. | L76 |
| 14 | Pls mention the range. | Okay, we added the range of about -6 to -1‰. | L94 |
| 15 | Please mention here the role of organic acids in chemical weathering of silicate rocks. | Okay, we added a sentence after the first sentence of this paragraph in L111: "In addition to carbonic acid, the main source of terrestrial weathering, organic acids from vegetation (mainly carboxylic acids) and strong inorganic acids (mainly sulphuric and nitric acid, derived from the oxidation of sulfides and ammonium, respectively; Raymond and Hamilton, 2018) drive the weathering reactions. However, to the best of our knowledge, there is not any study showing the effect of DOC on the weathering reaction" We also mentioned this in the sentence in L477: "Based on the $\delta^{13}$C-DIC signal, silicate bedrock is dissolved by sulphuric acid, and potentially organic acids, with no generation of DIC." | L114 |
| 16 | forcings | Okay, we changed this. | L146 |
| 17 | potential factors? | Okay, we changed this. | L147 |

| 18 | Pls give reference. | We added two references and revised this sentence to: "The Finnmarksvidda plateau was completely covered by the Fennoscandian ice sheet during the last glaciation (Olsen et al., 2013). In addition to being covered by the Fennoscandian ice sheet with an ice dome zone over Finland during the last stadial, the Finnmarksvidda was also covered by the Scandinavian ice sheets which grew from the mountainous area of northwest Sweden and from centers along the Caledonian mountain range in Norway during the Middle and Early Weichselian (Olsen, 1988; Olsen et al., 2013)." | L156 |
|---|---|---|---|
| 19 | Why the range is reverse? | We wanted to give the earlier date first. | L158 |
| 20 | Pls mention the dates. | That would be nine dates. In our opinion, this listing would disturb the reading flow. All data can be found in the Supplementary File under https://doi.pangaea.de/10.1594/PANGAEA.952905. We added a reference to the PANGAEA data set in line 201: "All data can be accessed at https://doi.org/10.1594/PANGAEA.952905". | L199 |
| 21 | Pls refer figure where ever is applicable for Ga, Gu etc. | Done! | L201 |
| 22 | Pls mention the concentrations of Meybeck (1987). | Okay, we added the concentration of 125 µmol L$^{-1}$ and added more information about the environmental conditions in the study by Meybeck (1987) and put it in context with our study: "…with more than 90% of the samples taken during non-flood periods. However, we would like to point out that the catchments studied by Meybeck (1987) are from temperate regions and ours are from Arctic regions." | L277 |
| 23 | Pls mention the range. | We added the range of 71-175 µmol L$^{-1}$. | L280 |
| 24 | A major concern. | Regarding the issue of not correcting our data for precipitation due to the lack of rainwater composition data at the study site, leading to the likely underestimation of the Ca$^{2+}$/Na$^{+}$ and Mg$^{2+}$/Na$^{+}$ molar ratios: We understand the reviewer's concern. However, we believe that we have shown, at least briefly in L284-L290, the consequences of not correcting our data for precipitation. We think that a precipitation correction would | L284 |

|  |  | not severely change the main conclusions of the investigations of the weathering pathways of the Gaskabohki headwater catchment, situated about 100 km away from the coastline, at an altitude of ~600 m amsl. We acknowledge the effect of sea spray on the composition of freshwater in coastal regions. However, we also recognize that our dataset does not include a reliable tool to quantitatively assess this effect. Additional complications arise from the effect that our sampling locations extend away from the coast, which would require spatial corrections in the corrections for sea spray. Therefore, we chose to provide upper bounds rather than introduce uncertainty that we cannot constrain. |  |